# Convolutional Normalization: Improving Deep Convolutional Network Robustness and Training

**Sheng Liu**[*]
New York University
shengliu@nyu.edu

**Xiao Li**
University of Michigan
xlxiao@umich.edu

**Yuexiang Zhai**
UC Berkeley
ysz@berkeley.edu

**Chong You**
Google Research
cyou@google.com

**Zhihui Zhu**
University of Denver
zhihui.zhu@du.edu

**Carlos Fernandez-Granda**
New York University
cfgranda@cims.nyu.edu

**Qing Qu**
University of Michigan
qingqu@umich.edu

## Abstract

Normalization techniques have become a basic component in modern convolutional neural networks (ConvNets). In particular, many recent works demonstrate that promoting the orthogonality of the weights helps train deep models and improve robustness. For ConvNets, most existing methods are based on penalizing or normalizing weight matrices derived from concatenating or flattening the convolutional kernels. These methods often destroy or ignore the benign convolutional structure of the kernels; therefore, they are often expensive or impractical for deep ConvNets. In contrast, we introduce a simple and efficient "Convolutional Normalization" (ConvNorm) method that can fully exploit the convolutional structure in the Fourier domain and serve as a simple plug-and-play module to be conveniently incorporated into any ConvNets. Our method is inspired by recent work on preconditioning methods for convolutional sparse coding and can effectively promote each layer's channel-wise isometry. Furthermore, we show that our ConvNorm can reduce the layerwise spectral norm of the weight matrices and hence improve the Lipschitzness of the network, leading to easier training and improved robustness for deep ConvNets. Applied to classification under noise corruptions and generative adversarial network (GAN), we show that the ConvNorm improves the robustness of common ConvNets such as ResNet and the performance of GAN. We verify our findings via numerical experiments on CIFAR and ImageNet. Our implementation is available online at https://github.com/shengliu66/ConvNorm.

## 1 Introduction

In the past decade, Convolutional Neural Networks (ConvNets) have achieved phenomenal success in many machine learning and computer vision applications [1–7]. Normalization is one of the most important components of modern network architectures [8]. Early normalization techniques, such as batch normalization (BatchNorm) [4], are cornerstones for effective training of models beyond a few layers. Since then, the values of normalization for optimization and learning is extensively studied, and many normalization techniques, such as layer normalization [9], instance normalization [10], and group normalization [11] are proposed. Many of such normalization techniques are based on estimating certain statistics of neuron inputs from training data. However, precise estimations of the statistics may not always be possible. For example, BatchNorm becomes ineffective when the batch size is small [12], or batch samples are statistically dependent [13].

---

[*]The first two authors contributed to this work equally.

35th Conference on Neural Information Processing Systems (NeurIPS 2021).

Weight normalization [14] is a powerful alternative to BatchNorm that improves the conditioning of neural network training without the need to estimate statistics from neuron inputs. Weight normalization operates by either reparameterizing or regularizing the network weights so that all the weights have unit Euclidean norm. Since then, various forms of normalization for network weights are proposed and become critical for many tasks such as training Generative Adversarial Networks (GANs) [15] and obtaining robustness to input perturbations [16, 17]. One of the most popular forms of weight normalization is enforcing orthogonality, which has drawn attention from a diverse range of research topics. The idea is that weights in each layer should be orthogonal and energy-preserving. Orthogonality is argued to play a central role for training ultra-deep models [18–22], optimizing recurrent models [23–26], improving generalization [27], obtaining robustness [28, 29], learning disentangled features [30, 31], improving the quality of GANs [32, 33], learning low-dimensional embedding [34], etc.

**Exploiting convolution structures for normalization.** Our work is motivated by the pivotal role of weight normalization in deep learning. In the context of ConvNets, the network weights are multi-dimensional (e.g., 4-dimensional for a 2D ConvNet) convolutional kernels. A vast majority of existing literature [27, 28, 35–39] imposes orthogonal weight regularization for ConvNets by treating multi-dimensional convolutional kernels as 2D matrices (e.g., by flattening certain dimensions) and imposing orthogonality of the matrix. However, this choice ignores the translation-invariance properties of convolutional operators and, as shown in [22], does not guarantee energy preservation. On the other hand, these methods often involve dealing with matrix inversions that are computationally expensive for deep and highly overparameterized networks.



Figure 1: **Comparison between BatchNorm and ConvNorm** on activations of $k = 1, \ldots, C$ channels. BatchNorm subtracts and multiplies the activations of each channel by computed scalars: mean $\mu$ and variance $\sigma^2$, before a per-channel affine transform parameterized by learned parameters $\beta$ and $\gamma$; ConvNorm performs per-channel convolution with precomputed kernel $v$ to normalize the spectrum of the weight matrix for the convolution layer, following with a channel-wise convolution with learned kernel $r$ as the affine transform.

In contrast, in this work we introduce a new normalization method dedicated to ConvNets, which explicitly exploits translation-invariance properties of convolutional operators. Therefore, we term our method as *Convolutional Normalization* (ConvNorm). We normalize each output channel for each layer of ConvNets, similar to recent preconditioning methods for convolutional sparse coding [40]. The ConvNorm can be viewed as a reparameterization approach for the kernels, that actually it normalizes the weight of each channel to be tight frame.[2] While extra mathematical hassles do exist in incorporating translation-invariance properties, and it turns out to be a blessing, rather than a curse, in terms of computation, as it allows us to carry out the inversion operation in our ConvNorm via fast Fourier transform (FFT) in the frequency domain, for which the computation complexity can be significantly reduced.

**Highlights of our method.** In summary, for ConvNets our approach enjoys several clear advantages over classical normalization methods [41–43], that we list below:

- **Easy to implement.** In contrast to weight regularization methods that often require hyperparameter tuning and heavy computation [41, 43], the ConvNorm has no parameter to tune and is efficient to compute. Moreover, the ConvNorm can serve as a simple plug-and-play module that can be conveniently incorporated into training almost any ConvNets.

- **Improving network robustness.** Although the ConvNorm operates on each output channel separately, we show that it actually improves the overall layer-wise Lipschitzness of the ConvNets. Therefore, as demonstrated by our experiments, it has superior robustness performance against noise corruptions and adversarial attacks.

- **Improving network training.** We numerically demonstrate that the ConvNorm accelerates training on standard image datasets such as CIFAR [44] and ImageNet [45]. Inspired by the work [40, 46], our high-level intuition is that the ConvNorm improves the optimization landscape that optimization algorithms converge faster to the desired solutions.

---

[2]Tight frame can be viewed as a generalization of orthogonality for overcomplete matrices, which is also energy preserving.

**Related work.** Besides our work, a few very recent work also exploits the translation-invariance for designing the normalization techniques of ConvNets. We summarize and explain the difference with our method below.

- The work [22, 43] derived a similar notion of orthogonality for convolutional kernels, and adopted a penalty based method to enforce orthogonality for network weights. These penalty methods often require careful tuning of the strength of the penalty on a case-by-case basis. In contrast, our method is parameter-free and thus easier to use. Our method also shows better empirical performance in terms of robustness.
- Very recent work by [29] presented a method to enforce *strict* orthogonality of convolutional weights by using Cayley transform. Like our approach, a sub-step of their method utilizes the idea of performing the computation in the Fourier domain. However, as they normalize the whole unstructured weight matrix, computing expensive matrix inversion is inevitable, so that their running time and memory consumption is prohibitive for large networks.[3] In contrast, our method is "orthogonalizing" the weight of each channel instead of the whole layer, so that we can exploit the convolutional structure to avoid expensive matrix inversion with a much lower computational burden. In the meanwhile, we show that this channel-wise normalization can still improve layer-wise Lipschitz condition.

**Organizations.** The rest of our paper is organized as follows. In Section 2, we introduce the basic notations and provide a brief overview of ConvNets. In Section 3, we introduce the design of the proposed ConvNorm and discuss the key intuitions and advantages. In Section 4, we perform extensive experiments on various applications verifying the effectiveness of the proposed method. Finally, we conclude and point to some interesting future directions in Section 5. To streamline our presentation, some technical details are deferred to the Appendices.

## 2 Preliminary

**Review of deep networks.** A deep network is essentially a *nonlinear* mapping $f(\cdot) : \boldsymbol{x} \mapsto \boldsymbol{y}$, which can be modeled by a composition of a series of simple maps: $f(\boldsymbol{x}) = f^{L-1} \circ \cdots \circ f^1 \circ f^0(\boldsymbol{x})$, where every $f^\ell(\cdot)$ ($1 \leq \ell \leq L$) is called one "layer". Each layer is composed of a linear transform, followed by a simple nonlinear activation function $\varphi(\cdot)$.[4] More precisely, a basic deep network of $L$ layers can be defined recursively by interleaving linear and nonlinear activation layers as

$$\boldsymbol{z}^{\ell+1} = f^\ell(\boldsymbol{z}^\ell) = \varphi \circ \mathcal{A}^\ell(\boldsymbol{z}^\ell) \tag{1}$$

for $\ell = 0, 1, \ldots, L-1$, with $\boldsymbol{z}_0 = \boldsymbol{x}$. Here $\mathcal{A}^\ell(\cdot)$ denotes the linear transform and will be described in detail soon. For convenience, let us use $\boldsymbol{\theta}$ to denote all network parameters in $\left\{\mathcal{A}^\ell(\cdot)\right\}_{\ell=0}^{L-1}$. The goal of deep learning is to fit the observation $\boldsymbol{y}$ with the output $f(\boldsymbol{x}, \boldsymbol{\theta})$ for any sample $\boldsymbol{x}$ from a distribution $\mathcal{D}$, by learning $\boldsymbol{\theta}$. This can be achieved by optimizing a certain loss function $\ell(\cdot)$, i.e.,

$$\min_{\boldsymbol{\theta} \in \boldsymbol{\Theta}} L\left(\boldsymbol{\theta}; \left\{\left(\boldsymbol{x}^i, \boldsymbol{y}^i\right)\right\}_{i=1}^m\right) := \frac{1}{m} \sum_{i=1}^m \ell\left(f(\boldsymbol{x}^i, \boldsymbol{\theta}), \boldsymbol{y}^i\right),$$

given a (large) training dataset $\left\{\left(\boldsymbol{x}^i, \boldsymbol{y}^i\right)\right\}_{i=1}^m$. For example, for a typical classification task, the class label of a sample $\boldsymbol{x}$ is represented by a one-hot vector $\boldsymbol{y} \in \mathbb{R}^k$ representing its membership in $k$ classes. The loss can be chosen to be either the cross-entropy or $\ell_2$-loss [48]. In the following, we use $(\boldsymbol{x}, \boldsymbol{y})$ to present one training sample.

**An overview of ConvNets.** The ConvNet [49] is a special deep network architecture, where each of its linear layer can be implemented much more efficiently via convolutions in comparison to fully connected networks [50]. Because of its efficiency and popularity in machine learning, for the rest of the paper, we focus on ConvNets. Suppose the input data $\boldsymbol{x}$ has $C$ channels, represented as

$$\boldsymbol{x} = (\boldsymbol{x}_1, \boldsymbol{x}_2, \cdots, \boldsymbol{x}_C), \tag{2}$$

where for 1D signal $\boldsymbol{x}_k \in \mathbb{R}^m$ denotes the $k$th channel feature of $\boldsymbol{x}$.[5] For the $\ell$th layer ($0 \leq \ell \leq L-1$) of ConvNets, the linear operator $\mathcal{A}^\ell(\cdot) : \mathbb{R}^{C_\ell \times m} \mapsto \mathbb{R}^{C_{\ell+1} \times m}$ in (1) is a convolution operation with $C_{\ell+1}$ output channels,

---

[3]In [29], the results are reported based on ResNet9, whereas our method can be easily added to larger networks, e.g. ResNet18 and ResNet50.

[4]The nonlinearity could contain BatchNorm [4], pooling, dropout [47], and stride, etc.

[5]If the data is 2D, we can assume $\boldsymbol{x} \in \mathbb{R}^{m_1 \times m_2}$. For simplicity, we present our idea based on 1D signal.

$$z^{\ell+1} = \left( z_1^{\ell+1}, z_2^{\ell+1}, \cdots, z_{C_{\ell+1}}^{\ell+1} \right),$$

$$z_k^{\ell+1} = \varphi \left( \sum_{j=1}^{C_\ell} a_{kj}^\ell * z_j^\ell \right) \quad (1 \le k \le C_{\ell+1}),$$

where $*$ denotes the convolution between two items that we will discuss below in more detail. Thus, for the $\ell$th layer with $C_\ell$ input channels and $C_{\ell+1}$ output channels, we can organize the convolution kernels $\{a_{kj}\}$ as

$$A^\ell = \begin{bmatrix} a_{11}^\ell & a_{12}^\ell & \cdots & a_{1C_\ell}^\ell \\ a_{21}^\ell & a_{22}^\ell & \cdots & a_{2C_\ell}^\ell \\ \vdots & \vdots & \ddots & \vdots \\ a_{C_{\ell+1}1}^\ell & a_{C_{\ell+1}2}^\ell & \cdots & a_{C_{\ell+1}C_\ell}^\ell \end{bmatrix}.$$

**Convolution operators.** For the simplicity of presentation and analysis, we adopt *circular* convolution instead of linear convolution.[6] For 1D signal, given a kernel $a \in \mathbb{R}^n$ and an input signal $x \in \mathbb{R}^m$ (in many cases $m \gg n$), a circular convolution $*$ between $a$ and $x$ can be written in a simple matrix-vector product form via

$$y = a * x = C_a \cdot x,$$

where $C_a$ denotes a circulant matrix of (zero-padded) $a$,

$$C_a := \begin{bmatrix} s_0[a] & s_1[a] & \cdots & s_{m-1}[a] \end{bmatrix},$$

which is the concatenation of all cyclic shifts $s_k[a]$ $(0 \le k \le m-1)$ of length $k$ of the (zero-padded) vector $a$. Since $C_a$ can be decomposed via the discrete Fourier transform (DFT) matrix $F$:

$$C_a = F^* \operatorname{diag}(\widehat{a}) F, \quad \widehat{a} = Fa, \tag{3}$$

where $\widehat{a}$ denotes the Fourier transform of a vector $a$. The computation of $a * x$ can be carried out efficiently via fast Fourier transform (FFT) in the frequency domain. We refer the readers to the appendix for more technical details.

# 3   Convolutional Normalization

In the following, we introduce the proposed ConvNorm, that can fully exploit benign convolution structures of ConvNets. It can be efficiently implemented in the frequency domain, and reduce the layer-wise Lipschitz constant. First of all, we build intuitions of the new design from the simplest setting. From this, we show how to expand the idea to practical ConvNets and discuss its advantages for training and robustness.

## 3.1   A warm-up study

Let us build some intuitions by zooming into one layer of ConvNets with both input and output being *single-channel*,

$$z_{out} = \mathcal{A}_L(z) = a * z_{in}, \tag{4}$$

where $z_{in}$ is the input signal, $a$ is a single kernel, and $z_{out}$ denotes the output before the nonlinear activation. The form (4) is closely related to recent work on blind deconvolution [46]. More specifically, the work showed that normalizing the output $z_{out}$ via *preconditioning* eliminates bad local minimizers and dramatically improves the optimization landscapes for learning the kernel $a$. The basic idea is to multiply a preconditioning matrix which approximates the following form[7]

$$P = \left( C_a C_a^\top \right)^{-1/2}. \tag{5}$$

As we observe

---

[6]Although there are slight differences between linear and circulant convolutions on the boundaries, actually any linear convolution can be *reduced* to circular convolution simply via zero-padding.

[7]In the work [46], they cook up a matrix by using output samples $\widetilde{P} = \left( \frac{C}{m} \sum_{i=1}^m C_{z_{out}^i} (C_{z_{out}^i})^\top \right)^{-1/2}$.
When the input samples $z_{in}^i$ are i.i.d. zero mean, it can be showed that $\widetilde{P} \approx P$ for large $m$. For ConvNets, we can just use the learned kernel $a$ for cooking up $P$.

$$\widetilde{z}_{out} \;=\; \boldsymbol{P} z_{out} \;=\; \underbrace{\left(\boldsymbol{C_a C_a^\top}\right)^{-1/2} \boldsymbol{C_a}}_{\boldsymbol{Q(a)}} \cdot z_{in},$$

the ConvNorm is essentially *reparametrizing* the circulant matrix $\boldsymbol{C_a}$ of the kernel $\boldsymbol{a}$ to an *orthogonal* circulant matrix $\boldsymbol{Q(a)} = \left(\boldsymbol{C_a C_a^\top}\right)^{-1/2} \boldsymbol{C_a}$, with $\boldsymbol{QQ^\top} = \boldsymbol{I}$. Thus, the ConvNorm is improving the conditioning of the *vanilla* problem and reducing the Lipschitz constant of the operator $\mathcal{A}_L(\cdot)$ in (4). On the other hand, the benefits of this normalization can also be observed in the frequency domain. Based on (3), we have $\boldsymbol{P} = \boldsymbol{F}^* \operatorname{diag}(\boldsymbol{v}) \boldsymbol{F} = \boldsymbol{C_v}$ with $\boldsymbol{v} = \boldsymbol{F}^{-1}\left(|\widehat{\boldsymbol{a}}|^{\odot -1}\right)$. Thus, we also have

$$\boldsymbol{Q(a)} = \boldsymbol{C_v} \cdot \boldsymbol{C_a} \;=\; \boldsymbol{C_{v*a}} = \boldsymbol{F}^* \operatorname{diag}(\widehat{\boldsymbol{g}}(\boldsymbol{a}))\boldsymbol{F}, \quad \widehat{\boldsymbol{g}}(\boldsymbol{a}) \;=\; \widehat{\boldsymbol{a}} \odot |\widehat{\boldsymbol{a}}|^{\odot -1},$$

with $\odot$ denoting entrywise operation and $\boldsymbol{g} = \boldsymbol{F}^{-1}\left(\widehat{\boldsymbol{a}} \odot |\widehat{\boldsymbol{a}}|^{\odot -1}\right)$. Thus, we can see that:

- Although the reparameterization involves matrix inversion, which is typically expensive to compute, for convolution it can actually be much more efficiently implemented in the frequency domain via FFT, reducing the complexity from $O(n^3)$ to $O(n \log n)$.
- The reparametrized kernel $\boldsymbol{g}$ is effectively an *all-pass* filter with flat normalized spectrum $\widehat{\boldsymbol{a}} \odot |\widehat{\boldsymbol{a}}|^{\odot -1}$.[8] From an information theory perspective, this implies that it can better preserve (in particular, high-frequency) information of the input feature from the previous layer.

### 3.2 ConvNorm for multiple channels

So far, we only considered one layer ConvNets with single-channel input and output. However, recall from Section 2, modern deep ConvNets are usually designed with many layers; each typical layer is constructed with a linear transformation with *multiple* input and output channels, followed by strides, normalization, and nonlinear activation. Extension of the normalization approach in Section 3.1 from one layer to multiple layers is easy, which can be done by applying the same normalization repetitively for all the layers. However, generalizing our method from a single channel to multiple channels is not obvious, that we discuss below.

In [40], the work introduced a preconditioning method for normalizing multiple kernels in convolutional sparse coding. In the following, we show that such an idea can be adapted to normalize each output channel, reduce the Lipschitz constant of the weight matrix in each layer, and improve training and network robustness. Let us consider any layer $\ell$ ($1 \leq \ell \leq L$) within a *vanilla* ConvNet using 1-stride, and take one channel (e.g., $k$-th channel) of that layer as an example. For simplicity of presentation, we hide the layer number $\ell$. Given $\boldsymbol{z}_{k,out} = \sum_{j=1}^{C_I} \boldsymbol{a}_{kj} * \boldsymbol{z}_{j,in}$, the $k$-th output channel can be written as

$$\boldsymbol{z}_{k,out} \;=\; \underbrace{\begin{bmatrix} \boldsymbol{C}_{\boldsymbol{a}_{k1}} & \boldsymbol{C}_{\boldsymbol{a}_{k2}} & \cdots & \boldsymbol{C}_{\boldsymbol{a}_{kC_I}} \end{bmatrix}}_{\boldsymbol{A}_k} \cdot \underbrace{\begin{bmatrix} \boldsymbol{z}_{1,in} \\ \boldsymbol{z}_{2,in} \\ \vdots \\ \boldsymbol{z}_{C_I,in} \end{bmatrix}}_{\boldsymbol{z}_{in}},$$

with $C_I$ and $C_O$ being the numbers of input and output channels, respectively. For each channel $k = 1, \cdots, C_O$, we normalize the output by

$$\boxed{\boldsymbol{P}_k \;=\; \left(\sum_{j=1}^{C_I} \boldsymbol{C}_{\boldsymbol{a}_{kj}} \boldsymbol{C}_{\boldsymbol{a}_{kj}}^\top\right)^{-1/2} = \left(\boldsymbol{A}_k \boldsymbol{A}_k^\top\right)^{-1/2},} \tag{6}$$

so that

$$\widetilde{\boldsymbol{z}}_{k,out} \;=\; \boldsymbol{P}_k \boldsymbol{z}_{k,out} \;=\; \underbrace{\left(\boldsymbol{A}_k \boldsymbol{A}_k^\top\right)^{-1/2} \boldsymbol{A}_k}_{\boldsymbol{Q}_k(\boldsymbol{A}_k)} \cdot \boldsymbol{z}_{in}. \tag{7}$$

Thus, we can see the ConvNorm is essentially a reparameterization of the kernels $\{\boldsymbol{a}_{kj}\}_{j=1}^{C_I}$ for the $k$-th channel. Similar to Section 3.1, the operation can be rewritten in the form of convolutions

$$\boldsymbol{Q}_k(\boldsymbol{A}_k) \;=\; \boldsymbol{P}_k \boldsymbol{A}_k \;=\; \begin{bmatrix} \boldsymbol{C}_{\boldsymbol{v}_k * \boldsymbol{a}_{k1}} & \cdots & \boldsymbol{C}_{\boldsymbol{v}_k * \boldsymbol{a}_{kC_I}} \end{bmatrix}$$

---

[8] An all-pass filter is a signal processing filter that passes all frequencies equally in gain, but can change the phase relationship among various frequencies.

with $P_k = C_{v_k}$ and $v_k = F^{-1}\left(\sum_{i=1}^{C_I} |\widehat{a}_{ki}|^{\odot 2}\right)^{\odot -1/2}$; it can be efficiently implemented via FFT.

Here, as for multiple kernels the matrix $A_k$ is *overcomplete* (i.e., $A_k$ is a wide rectangular matrix), we *cannot* normalize the channel-wise weight matrix $A_k$ to exact orthogonal. However, it can be normalized to *tight frame* with $Q_k Q_k^\top = I$. This further implies that we can normalize the spectral norm $\|Q_k\|$ of the weight matrix $Q_k$ in each channel to unity (see Figure 2 (Left)).

Combining the operation for all the channels, the ConvNorm for each layer overall can be summarized as follows:

$$\widetilde{z}_{out} = \begin{bmatrix} P_1 z_{1,out} \\ \vdots \\ P_{C_O} z_{C_O,out} \end{bmatrix} = \underbrace{\begin{bmatrix} Q_1 \\ \vdots \\ Q_{C_O} \end{bmatrix}}_{Q} z_{in}, \quad (8)$$

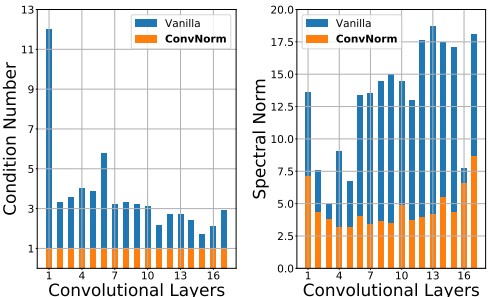

Figure 2: **Condition number for each channel (averaged over all channels) (Left), and spectral norm for each layer (Right)** on ResNet18 except for skip connection layers. ConvNorm normalizes the channel-wise condition number to 1 and reduces the layer-wise spectral norm. We use the method in [51] to calculate the singular values of the weight matrix.

that we normalize each output channel $k$ by different matrix $P_k$. The proposed ConvNorm has several advantages that we discuss below.

**Proposition 3.1** *The spectral norm of $Q$ introduced in* (8) *can be bounded by*

$$\|Q\| \leq \sqrt{\sum_{k=1}^{C_O} \|Q_k\|^2},$$

*that spectral norm of $Q$ is bounded by the spectral norms of all the weights $\{Q_k\}_{k=1}^{C_O}$.*

**Proof** We defer the proof to the Appendix A.3. ∎

- **Efficient implementations.** There are many existing results [29, 39, 41] trying to normalize the whole layerwise weight matrix. For ConvNets, as the matrix is neither circulant nor block circulant, computing its inversion is often computationally prohibitive. Here, for each layer, we only normalize the weight matrix of the individual output channel. Thus similar to Section 3.1, the inversion in (6) can be much more efficiently computed via FFT by exploiting the benign convolutional structure.
- **Improving layer-wise Lipschitzness.** As we can see from Proposition 3.1, although ConvNorm only normalized the spectral norm of each channel, it can actually reduce the spectral norm of the whole weight matrix, improving the Lipschitzness of each layer; see Figure 2 (Right) for a numerical demonstration on ResNet18. As extensively investigated [28, 29, 42], improving the Lipschitzness of the weights for ConvNets will lead to enhanced robustness against data corruptions, for which we will demonstrate on the proposed ConvNorm in Section 4.1.
- **Easier training and better generalization.** For deconvolution and convolutional sparse coding problems, the work [40, 46] showed that ConvNorm could dramatically improve the corresponding nonconvex optimization landscapes. On the other hand, from an algorithmic unrolling perspective for neural network design [52, 53], the ConvNorm is analogous to the preconditioned or conjugate gradient methods [54] which often substantially boost algorithmic convergence. Therefore, we conjecture that the ConvNorm also leads to better optimization landscapes for training ConvNets, that they can be optimized faster to better solution qualities of generalization. We empirically show this in Section 4.2.

### 3.3 Extra technical details

To achieve the full performance and efficiency potentials of the proposed ConvNorm, we discuss some essential implementation details in the following.

**Efficient back-propagation.** For ConvNorm, as the normalization matrix in (6) is constructed from the learned kernels, it complicates the computation of the gradient in back-propagation when training the network. Fortunately, we observe that treating the normalization matrices $\{P_k\}$ as constants during back-propagation usually does *not* affect the training and generalization performances, so that the computational complexity in training is not increased. We noticed that such a technique has also been recently considered in [55] for self-supervised learning, which is termed as *stop-gradient*.

**Learnable affine tranform.** For each channel, we include an (optional) *affine transform* after the normalization $P_k \cdot z_{k,out} = C_{v_k} \cdot z_{k,out} = v_k * z_{k,out}$ in (7) as follows:

$$\overline{z}_k \;=\; r_k * \widetilde{z}_{k,out} \;=\; r_k * v_k * z_{k,out},$$

where the extra convolutional kernel $r_k$ is learned along with the original model parameters. The idea of including this affine transform is analogous to including a learnable rescaling in BatchNorm, which can be considered as an "undo" operation to make sure the identity transform can be represented [4]. The difference between our affine transform and BatchNorm is that we apply channel-wise convolutions instead of simple rescaling (see Figure 1). Note that when $r_k$ is an inverse kernel of $v_k$ (i.e., $r_k * v_k = 1$), the overall transformation becomes an identity. The effectiveness of affine transform is demonstrated in the ablation study in Appendix C.4.

**Dealing with stride and 2D convolution.** There are extra technicalities that we briefly discuss below. For more details, we refer the readers to Appendix B.

- *Extension to 2D convolution.* Although we introduced the ConvNorm based on 1D convolution for the simplicity of presentations, it should be noted that our approach can be easily extended to the 2D case via 2D FFT.
- *Dealing with stride.* Strided convolutions are universal in modern ConvNet architectures such as the ResNet [5], which can be viewed as downsampling after unstrided convolution. To deal with stride for our ConvNorm, we first perform an unstrided convolution, normalizing the activations using ConvNorm and then downsampling the normalized activations. In comparison, the method proposed in [29] is incompatible with strided convolutions.

## 4  Experiments & Results

In this section, we run extensive experiments on CIFAR and ImageNet, empirically demonstrating two major advantages of our approach: *(i)* it improves the *robustness* against adversarial attacks, data scarcity, and label noise corruptions [56–58], and *(ii)* it makes deep ConvNets *easier to train* and perform better on problems such as classification and GANs [59]. The rest of this section is organized as follows. First, we introduce baseline methods for comparisons, and describe the setups of network architectures, datasets, and training. In Section 4.1 and Section 4.2, we demonstrate the effectiveness of our approach on robustness and training, respectively.

**Baseline methods for comparisons.** We compare our method with three representative methods.

- **Spectral normalization (SN).** For each layer of ConvNets, the work [15] treats multi-dimensional convolutional kernels as 2D matrices (e.g., by flattening certain dimensions) and normalizes its spectrum (i.e., singular values). It estimates the matrix's maximum singular value via a power method and then uses it to normalize all the singular values. As we discussed in Section 1, the method does not exploit convolutional structures of ConvNets.

- **Orthogonalization by Newton's Iteration (ONI).** The work [39] whitens the same reshaped matrices as SN, so that the reshaped matrices are reparametrized to orthogonality. However, the method needs to compute full inversions of covariance matrices, which is approximated by Newton's iterations. Again, no convolutional structure is utilized.

- **Orthogonal ConvNets (OCNN).** Few methods that exploit convolutional structures are [22, 43], which enforce orthogonality on doubly block circulant matrices of kernels via penalties on the loss. Here, we compare with [43].

**Setups of dataset, network and training.** For all experiments, if not otherwise mentioned, CIFAR-10 and CIFAR-100 datasets are processed with standard augmentations, i.e., random cropping and flipping. We use 10% of the training set for validation and treat the validation set as a held-out test set. For ImageNet, we perform standard random resizing and flipping. For training, we observe our ConvNorm is not sensitive to the learning rate, and thus we fix the initial learning rate to 0.1 for all experiments.[9] For experiments on CIFAR-10, we run 120 epochs and divide the learning rate by 10 at the 40th and 80th epochs; for CIFAR-100, we run 150 epochs and divide the learning rate by 10 at the 60th and 120th epoch; for ImageNet,we run 90 epochs and divide the learning rate by 10 at the 30th and 90th epochs. The optimization is done using SGD with a momentum of 0.9 and a weight decay of 0.0001 for all datasets. For networks we use two backbone networks: VGG16 [60] and ResNet18 [5]. We adopt Xavier uniform initialization [61] which is the default initialization in PyTorch for all networks.

---

[9]For experiments with ONI, we use learning rate 0.01 since the loss would be trained to NaN if with 0.1.

| $\epsilon$ | Test Acc. | SN | BN | ONI | OCNN | ConvNorm |
|---|---|---|---|---|---|---|
| 0 | Clean | $82.52 \pm 0.22$ | $82.13 \pm 0.67$ | $80.70 \pm 0.14$ | $82.90 \pm 0.31$ | $\mathbf{83.23 \pm 0.25}$ |
| $\frac{8}{255}$ | FGSM | $52.34 \pm 0.33$ | $51.72 \pm 0.52$ | $48.33 \pm 0.16$ | $52.49 \pm 0.21$ | $\mathbf{52.87 \pm 0.24}$ |
| | PGD-10 | $45.68 \pm 0.40$ | $45.31 \pm 0.29$ | $42.30 \pm 0.24$ | $45.74 \pm 0.13$ | $\mathbf{46.12 \pm 0.26}$ |
| | PGD-20 | $44.47 \pm 0.37$ | $44.04 \pm 0.24$ | $41.08 \pm 0.30$ | $44.53 \pm 0.10$ | $\mathbf{44.75 \pm 0.30}$ |

Table 1: **Comparison of ConvNorm to baseline methods under different gradient based attacks.** Models are robustly trained following the procedure in [62] using a ResNet18 backbone. Experiments are conducted on CIFAR-10 dataset. Results are averaged over 4 random seeds.

## 4.1 Improved robustness

In this section, we demonstrate our method is more robust to various kinds of adversarial attacks, as well as random label corruptions and small training datasets.

**Robustness against adversarial attacks.** Existing results [29, 43] show that controlling the layer-wise Lipschitz constants for deep networks improves robustness against adversarial attack. Since our method improves the Lipschitzness of weights (see Figure 2), we demonstrate its robustness under adversarial attack on the CIFAR-10 dataset. We adopt both white-box (gradient based) attack [57, 58] and black-box attack [56] to test the robustness of our proposed method and other baseline methods. The results are presented in Table 1 and Table 2. For the ease of presentation, all technical details about model training and generation of the adversarial examples are postponed to Appendix C.

| Method | Average Queries | Attack Success rate (%) |
|---|---|---|
| SN | 2519.32 | 60.60 |
| ONI | 2817.09 | 55.90 |
| OCNN | 2892.81 | 54.50 |
| ConvNorm | **2966.16** | **53.50** |

Table 2: **Comparison of ConvNorm to baseline methods on SimBA black box attack.** The mean value of average queries (the higher, the better) and attack success rate (the lower, the better) throughout 3 runs are reported. Models are trained using a ResNet18 backbone without BN layers.

In the case of gradient based attacks, we follow the training procedure described in [62] to train models with our ConvNorm and other baseline methods. We report the performances of the robustly trained models on both the clean test dataset and datasets that are perturbed by Fast Gradient Sign Method (FGSM) [57] and Projected Gradient Method (PGD) [58]. As shown in Table 1, our ConvNorm outperforms other methods in terms of robustness under white-box attack while maintaining a good performance on clean test accuracy.

For black-box attack, we adopt a popular black-box adversarial attack method, Simple Black-box Adversarial Attacks (SimBA) [56]. By submitting queries to a model for updated test accuracy, the attack method iteratively finds a perturbation where the confidence score drops the most. We report the average queries and success rate after 3072 iterations in Table 2. As we can see, the ConvNorm resists the most queries, and that the SimBA has the lowest attack success rate for ConvNorm compared with other baseline methods.

**Robustness against label noise and data scarcity.** It has been widely observed that overparameterized ConvNets tend to overfit when label noise presents or the amount of training labels is limited [63–66]. Recent work [67] shows that normalizing the weights enforces certain regularizations, which can improve generalization performance against both label noise and data scarcity. Since our method is essentially reparametrizing and normalizing the weights, we demonstrate the robustness of our approach under these settings on CIFAR-10 with ResNet18 backbone.

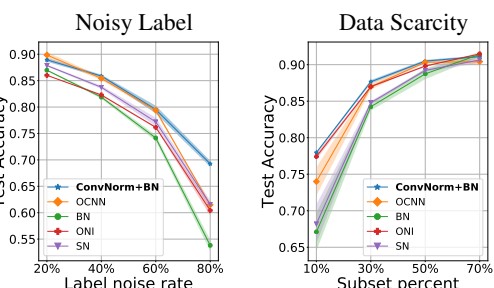

Figure 3: **Test accuracy for noisy label (Left) and insufficient training data (Right)**. Experiments are conducted on CIFAR-10 dataset using a ResNet18 backbone. Error bars corresponding to standard deviations over 3 runs.

- *Robustness against label noise.* Following the scheme proposed in [68], we simulate noisy labels by randomly flipping 20% to 80% of the labels in the training set. As shown in Figure 3 (Left), our method outperforms the others on most noisy rates by a hefty margin when the noise level is high.

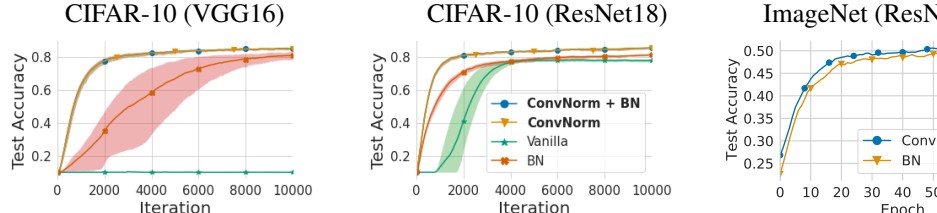

Figure 4: **ConvNorm accelerates convergence.** VGG16 trained on CIFAR-10 (Left), and ResNet18 (Middle) trained on CIFAR-10 and ImageNet (Right), with and without ConvNorm or BatchNorm. We do not use data augmentation, weight decay, or any other regularization in this experiment to isolate the effects of the normalization techniques. Error bars correspond to min/max over 4 runs.

- *Robustness against data scarcity.* We test our method on training the network with varying sizes of the training set, obtained by randomly sampling. The results in Figure 3 (Right) show that our ConvNorm achieves on par performance compared with baseline methods, and its performance stays high even when the size of the training data is tiny (e.g., 4500 examples).

## 4.2 Easier training on classification and GAN

Finally, we compare training convergence speed for classification and performances on GAN. Extra experiments on better generalization performance and ablation study can be found in Appendix C.4.

**Improved training on supervised learning.** We test our method on image classification tasks with two backbone architectures: VGG16 and ResNet18. We show that ConvNorm accelerates the convergence of training. To isolate the effects of the normalization layers for training, we train on CIFAR-10 and ImageNet without using any augmentation, regularization, and learning rate decay. In Figure 4, we show that adding ConvNorm consistently results in faster convergence, stable training (less variance in accuracy), and superior performance. On CIFAR-10, there is a wide performance gap after the first few iterations of training: 1000 iterations of training with ConvNorm lead to generalization performance comparable to 8000 iterations of training using BatchNorm. In the case of standard settings where data augmentation, regularization and learning rate decay are added, we notice that using ConvNorm and BatchNorm together also yield better test performances compared to only using BatchNorm (See Appendix C.4 for details). Besides the convergence speed of training, the exact training time for different methods is another important factor for measuring the efficiency of such methods. To this end, we empirically compare the training time for different methods and report the results in Appendix D and Table 8.

**Improved performance for GANs.** It has been found that improving the Lipschitz condition of the discriminator of GAN stabilizes its training [69]. For instance, WGAN-GP [70] demonstrates that adding a gradient penalty (1-GP) regularization to enforce the 1-Lipschitzness of the discriminator stabilizes GAN training and prevents mode collapse. Subsequent works [71, 72] using variants of the 1-GP regularization also show their improvement in GAN. Later on, [15] further reveals the performance of GAN can be significantly improved if the spectral norm (Lipschitz condition) of the discriminator network is strictly enforced to 1. As shown in Figure 2, the proposed ConvNorm also controls the Lipschitz condition of ConvNets. Therefore, we expect our method to also ameliorates the performance of GAN.

| Metric | SN | ONI | OCNN | Vanilla | ConvNorm |
|--------|------|-------|-------|---------|----------|
| IS | **8.12** | 7.07 | 7.54 | 7.13 | 7.62 |
| FID | **14.53** | 29.49 | 22.15 | 29.47 | 21.37 |

To demonstrate the effectiveness of the ConvNorm on GAN, we compare it with other baseline methods introduced previously. In our experiments, we adopt the same settings and architecture suggested in [15] without any modification, and we use the inception score (IS) [73], and FID [74] score for quantitative evaluation. As shown in Table 3, our ConvNorm achieves the second-best performance to SN.[10]

Table 3: **Comparison of ConvNorm to baseline methods on GAN training.** Inception score (IS) (the higher, the better) and FID score (the lower, the better) of ResNet with different normalizations. For each pair of model and method, we generate $50k$ images 10 times and compute the mean of IS.

---

[10]The performance of GANs is highly sensitive to the computational budget and the hyperparameters of the networks [75], and the hyperparameters of SN is fine-tuned for CIFAR-10 while we use the same hyperparameters as SN.

# 5    Discussions & Conclusion

In this work, we introduced a new normalization approach for ConvNets, which explicitly exploits translation-invariance properties of convolutional operators, leading to efficient implementation and boosted performances in training, generalization, and robustness. Our work has opened several interesting directions to be further exploited for normalization design of ConvNets: *(i)* although we provided some high-level intuitions why our ConvNorm works, theoretical justifications are needed; *(ii)* as our ConvNorm only promotes channel-wise "orthogonality", it would be interesting to utilize similar ideas to efficiently normalize the layerwise weight matrices by exploiting convolutional structures. We leave these questions for future investigations.

## Acknowledgement

Part of this work was done when XL and QQ were at Center for Data Science, NYU. SL, XL, CFG, and QQ were partially supported by NSF grant DMS 2009752. SL was partially supported by NSF NRT-HDR Award 1922658. CY acknowledges support from Tsinghua-Berkeley Shenzhen Institute Research Fund. ZZ acknowledges support from NSF grant CCF 2008460. QQ also acknowledges support of Moore-Sloan fellowship, and startup fund at the University of Michigan.

