# Appendices

The whole appendix is organized as follows.

- In Appendix A, we introduce the basic notations that are used throughout the paper and the appendix, and introduce the basic tools for analysis.

- In Appendix B, we describe the implementation details of our ConvNorm, including details for dealing with 2D convolutions, strides, paddings, and the discuss about the differences between different types of convolutions.

- In Appendix C, we describe the the experimental settings for Section 4 in detail.

- Finally, in Appendix D we conduct a more comprehensive ablation study on the influence of different components of the proposed ConvNorm.

## A   Notations & basic tools

### A.1   Notations

Throughout this paper, all vectors/matrices are written in bold font $\boldsymbol{a}/\boldsymbol{A}$; indexed values are written as $a_i, A_{ij}$. For a matrix $\boldsymbol{A} \in \mathbb{C}^{m \times n}$, we use $\boldsymbol{A}^\top$ and $\boldsymbol{A}^*$ to denote the transpose and conjugate transpose of $\boldsymbol{A}$, respectively. We let $[m] = \{1, 2, \cdots, m\}$. Let $\boldsymbol{F}_n \in \mathbb{C}^{n \times n}$ denote a unnormalized $n \times n$ DFT matrix, with $\|\boldsymbol{F}_n\| = \sqrt{n}$, and $\boldsymbol{F}_n^{-1} = n^{-1}\boldsymbol{F}_n^*$. In many cases, we just use $\boldsymbol{F}$ to denote the DFT matrix. For any vector $\boldsymbol{v} \in \mathbb{C}^n$, we use $\widehat{\boldsymbol{v}} = \boldsymbol{F}\boldsymbol{v}$ to denote its Fourier transform, and $\overline{\boldsymbol{v}}$ denotes the conjugate of $\boldsymbol{v}$. We use $*$ to denote the *circular* convolution with modulo-$n$: $(\boldsymbol{v} * \boldsymbol{u})_i = \sum_{j=0}^{m-1} v_j u_{i-j}$, and we use $\circledast$ to denote the cross-correlation $\boldsymbol{v} \circledast \boldsymbol{u}$ used in modern ConvNets.

### A.2   Circular convolution and circulant matrices.

For a vector $\boldsymbol{v} \in \mathbb{R}^n$, let $\mathrm{s}_\ell[\boldsymbol{v}]$ denote the cyclic shift of $\boldsymbol{v}$ with length $\ell$. Thus, we can introduce the circulant matrix $\boldsymbol{C_v} \in \mathbb{R}^{n \times n}$ generated through $\boldsymbol{v} \in \mathbb{R}^n$, that is,

$$\boldsymbol{C_v} = \begin{bmatrix} v_1 & v_n & \cdots & v_3 & v_2 \\ v_2 & v_1 & v_n & & v_3 \\ \vdots & v_2 & v_1 & \ddots & \vdots \\ v_{n-1} & & \ddots & \ddots & v_n \\ v_n & v_{n-1} & \cdots & v_2 & v_1 \end{bmatrix} = \begin{bmatrix} \mathrm{s}_0\left[\boldsymbol{v}\right] & \mathrm{s}_1\left[\boldsymbol{v}\right] & \cdots & \mathrm{s}_{n-1}\left[\boldsymbol{v}\right] \end{bmatrix}.$$

Now the circular convolution can also be written in a simpler matrix-vector product form. For instance, for any $\boldsymbol{u}, \boldsymbol{v} \in \mathbb{R}^n$, we have

$$\boldsymbol{u} * \boldsymbol{v} = \boldsymbol{C_u} \cdot \boldsymbol{v} = \boldsymbol{C_v} \cdot \boldsymbol{u}.$$

In addition, the cross-correlation between $\boldsymbol{u}$ and $\boldsymbol{v}$ can be also written in a similar form of convolution operator which reverses one vector before convolution with $\check{\boldsymbol{u}} * \boldsymbol{v}$, where $\check{\boldsymbol{v}}$ denote a *cyclic reversal* of $\boldsymbol{v} \in \mathbb{R}^m$ (i.e., $\check{\boldsymbol{v}} = [v_1, v_m, v_{m-1}, \cdots, v_2]^\top$).

### A.3   Proof of Proposition 3.1

We restate Proposition 3.1 in Section 3 in the following.

**Proposition A.1** *The spectral norm of $\boldsymbol{Q}$ introduced in (8) can be bounded by*

$$\|\boldsymbol{Q}\| \leq \sqrt{\sum_{k=1}^{C_O} \|\boldsymbol{Q}_k\|^2},$$

*that spectral norm of $\boldsymbol{Q}$ is bounded by the spectral norms of all the weights $\{\boldsymbol{Q}_k\}_{k=1}^{C_O}$.*

**Proof** Suppose we have a matrix of the form $\boldsymbol{Q} = \begin{bmatrix} \boldsymbol{Q}_1 \\ \vdots \\ \boldsymbol{Q}_{C_O} \end{bmatrix}$, then by using the relationship between singular values and eigenvalues,

$$\sigma_1^2(\boldsymbol{Q}) \; = \; \lambda_1\left(\boldsymbol{Q}^\top \boldsymbol{Q}\right) \; = \; \lambda_1\left(\sum_{i=1}^{C_O} \boldsymbol{Q}_i^\top \boldsymbol{Q}_i\right) \; \leq \; \sum_{i=1}^{C_O} \lambda_1\left(\boldsymbol{Q}_i^\top \boldsymbol{Q}_i\right) = \sum_{i=1}^{C_O} \sigma_1^2(\boldsymbol{Q}_i).$$

Thus, we have

$$\sigma_1(\boldsymbol{Q}) \; \leq \; \sqrt{\sum_{i=1}^{C_O} \sigma_1^2(\boldsymbol{Q}_i)},$$

as desired. ■

## B  Implementation details for Section 3

In the main paper, for the ease of presentation we only introduced and discussed the high-level idea of the proposed ConvNorm, with few technical details missing. Here, we discuss the implementation details of the ConvNorm for ConvNets in practice. More specifically, Appendix B.1 provides the pseudocode of ConvNorm with circular convolutions, which is easy for presentation and analysis. It should be noted that modern ConvNets often use cross-correlation instead of circular convolutions. Hence in Appendix B.2 and Appendix B.3, we discuss in detail on how to deal with this difference in practice. Additionally, in Appendix B.4 and Appendix B.5, we include other implementation details, such as dealing with strides, and extensions from 1D to 2D convolutions.

### B.1  Algorithms

First of all, in Algorithm 1 we provide detailed pseudocode of implementing the proposed ConvNorm in ConvNets for 2D input data, where the convolution operations are based on circular convolutions. From our discussion in Section 3 , we can see that all the operations can be efficiently implemented in the frequency domain via 2D FFTs.[11]

It should be noted that modern ConvNets often use cross-correlation rather than the circular convolution. Nonetheless, we discuss the differences and similarities between the two in the following. Based on this, we show how to adapt Algorithm 1 to modern ConvNets (see Appendix B.2).

### B.2  Dealing with convolutions in ConvNets

Throughout the main body of the work, our description and analysis of ConvNorm are based on circular convolutions for the simplicity of presentations. However, it should be noted that current ConvNets typically use *cross correlation* in each convolutional layer, which can be viewed as a variant of the classical linear convolution with flipped kernels. Hence, to adapt our analysis from circular convolution to cross-correlation (i.e., the typical convolution used in ConvNets), we need to build some sense of "equivalence"between them. Since linear convolution has a close connection with both, we use linear convolution as a bridge to introduce the relationship and thus find the "equivalence" between circular convolution and cross-correlation. Based on this, we show how to adapt from circular convolution in Algorithm 1 to the convolution used in modern ConvNets by simple modifications.

**Relationship among all convolutions.**   In the following presentations, assume we have a signal $\boldsymbol{x} \in \mathbb{R}^n$ and a kernel vector $\boldsymbol{a} \in \mathbb{R}^m$ with $m \leq n$. We first discuss the connections between linear convolution and the other two types of convolutions, and then establish the equivalence between circular convolution and cross-correlation upon the observed connections. Figure 5 demonstrates a simple example of the connections.

---

[11]During evaluation, we use the moving average of $\widehat{\boldsymbol{v}}_k$ during training, the momentum of the moving average is obtained by a cosine rampdown function $0.5\left(1 + \cos\left(\frac{\min(\text{iter}, 40000)}{40000}\pi\right)\right)$, where iter is the current iteration.

---

**Algorithm 1** Pseudocode of the proposed **ConvNorm** in each layer of ConvNets with 2D inputs.

---

**Require:** $z_{out} \in \mathbb{R}^{B \times C_O \times W \times H}$    = convolution outputs with batchsize $B$, channels $C_O$, width $W$, and height $H$
**Require:** $a \in \mathbb{R}^{C_O \times C_I \times k_1 \times k_2}$    = kernels for all output channels $C_O$, input channels $C_I$, and kernel size $k_1 \times k_2$
**Require:** $r \in \mathbb{R}^{C_O \times k_1 \times k_2}$    = $C_O$ trainable kernels for affine transform with the same size of $a$
**for** $k$ in $[1, \ldots, C_O]$    ▷ for each output channel
**do**
    $\widehat{z}_{k,out} \leftarrow \mathrm{FFT}(z_{k,out})$    ▷ apply 2D Fast Fourier Transform (FFT) on convolution output
    $\widehat{a}_k \leftarrow \mathrm{FFT}(a_k)$    ▷ apply 2D FFT on kernels
    $\widehat{a}_k \leftarrow \mathtt{stop\_gradient}(\widehat{a}_k)$    ▷ treating $\widehat{a}_k$ as constants during back-propagation
    $\widehat{v}_k \leftarrow \left( \sum_{i=1}^{C_I} |\widehat{a}_{ki}|^{\odot 2} \right)^{\odot -1/2}$    ▷ this is the 2D FFT of $v_k$
    $\widetilde{z}_{k,out} \leftarrow \mathrm{IFFT}\left( \widehat{z}_{k,out} \odot \widehat{v}_k \right)$    ▷ circularly convolve $z_{out,k}$ with $v_k$
    $\bar{z}_{k,out} \leftarrow r_k * \widetilde{z}_{k,out}$    ▷ learnable affine transformation with $r_k$
**endfor**
**return** $\bar{z}_{k,out}$    ▷ normalized convolution output

---

- **Linear convolution & circular convolution.** The (finite, discrete) linear and circular convolution can both be written as

$$y(k) = \sum_{j=0}^{n-1} a(k-j) x(j).$$

  Despite the same written form, they differ in two ways: length and index. As illustrated in Figure 5 (i), linear convolution doesn't have constraints on the input length and the result always has length $n + m - 1$. In comparison, circular convolution requires both the kernel $a$ and the signal $x$ to share the same length. Therefore, as in Figure 5 (iii) and (iv), we always reduce linear convolution to circular convolution by zero padding both the kernel and the signal to the same length $n + m - 1$, as shown in the example in Figure 5 (iv). It should be noted that the reason for such length difference is also rooted in their different indexing methods. In the case of linear convolution, when indices fall outside the defined regions, the associated entries are 0, e.g., $a(-1) = 0$ and $x(3) = 0$ as shown Figure 5 (i). On the other hand, circular convolution uses the periodic indexing method, i.e., $a(-j) = a(m-j)$. For example, in Figure 5 (iii), $a(-2) = a(3-2) = 4$.

- **Linear convolution & cross-correlation.** As shown in Figure 5 (i) and (ii), both linear convolution and cross-correlation operations apply the so-called *sliding window* of the kernel $a$ to the signal $x$, where the sliding window moves to the right one step at a time when the stride equals one. However, notice that cross-correlation uses a flipped kernel compared with linear convolution. Another difference is in the length of the output, where the output of a linear convolution is of length $n + m - 1$, while the output of cross-correlation is of length $n - m + 1$. This is due to the fact that the cross-correlation operation does not calculate outputs for out-of-region indices (see the difference between Figure 5 (i) and (ii) for an example). To sum up, a cross-correlation is equivalent to a kernel-flipped and truncated linear convolution. Moreover, the amount of truncation is controlled by the amount of zero-padding on the signal $x$ in cross-correlation. For example, in Figure 5 (ii), there is no zero-padding and hence the result is equivalent to truncate the first and last elements from the result in Figure 5 (i); but consider if we zero-pad the signal $x$ by 1 element on both sides in Figure 5 (ii), then the result would be identical with Figure 5 (i). In general, we found that if we zero-pad the signal $x$ by $m - 1$ elements on both sides, a cross-correlation is equivalent to a kernel-flipped linear convolution without any truncation. We will discuss more about dealing with zero-padding in Appendix B.3.

**Adapting circular convolution to cross-correlation in ConvNets.**    Thus, based on these connections discussed above, we could now establish the "equivalence" between circular convolution and cross-correlation based on their connections to linear convolution, and hence adapt the proposed ConvNorm in Algorithm 1 with cross-correlations in ConvNets via the following steps:

1. Zero pad both sides of $z_{in}$ by $m - 1$ elements to get $\dot{z}_{in}$.
2. Perform the cross-correlation between the kernel $a$ and the input $\dot{z}_{in}$ to obtain the output $z_{out}$.

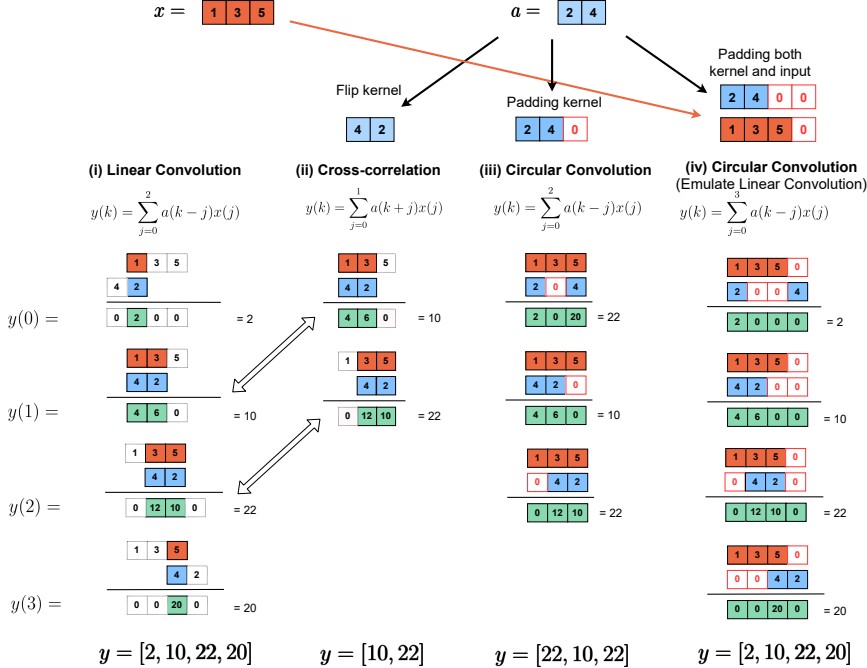

Figure 5: **A 1D example to illustrate relationships among different kinds of convolutions.** (i), (ii), and (iii) show the operations of linear convolution, cross-correlation (i.e., the "convolution" used in ConvNets), and circular convolution, respectively. (iv) gives an example of emulating linear convolution in a circular convolution manner. (i), (ii) indicates that cross-correlation is essentially a linear convolution with a flipped-kernel and truncation. Hence from (i), (ii), (iv), we find equivalence between cross-correlation and circular convolution.

3. Apply ConvNorm on the kernel $a$ and the output $z_{out}$ as stated in Algorithm 1 to get result $\widetilde{z}_{out}$.

4. Delete the first and last $(m-1)/2$ elements of $\widetilde{z}_{out}$ and return as the resulting output.

Here, Step 1 and Step 2 are to generate the output $z_{out}$ that is almost identical to what used in ConvNets, with the exception of zero-padding in Step 1 so that it is equivalent to a circular convolution with a flipped kernel $\breve{a}$. Then in Step 3, we perform ConvNorm on the output $z_{out}$ and kernel $a$. Notice that there is no need to flip the kernel in the above steps since as described in Algorithm 1, we only need to calculate the magnitude spectrum of a kernel and the magnitude spectrum remains consistent with a kernel flipping, i.e., $|Fa| = |F\breve{a}|$. Finally, Step 4 is to obtain the desired output with the correct spatial dimension.

## B.3    Dealing with zero-paddings in ConvNets

Here, we provide more explanations about the zero-padding and truncation used in Appendix B.2. Zero-padding is an operation of adding 0s to the data, which is widely used in modern ConvNets primarily aimed for maintaining the spatial dimension of the outputs for each layer. For example, a standard stride-1 convolution in ConvNets between a kernel $a \in \mathbb{R}^m$ and a signal $z_{in} \in \mathbb{R}^n$ with $n > m$ produces a output vector $z_{out}$ of length $n - m + 1$. To make the output $z_{out}$ the same length as the input signal $z_{in}$, a zero-padding of $\lfloor m/2 \rfloor$ is often used (e.g., common in various architectures such as VGG [60], ResNet [5].[12] To handle with such zero-padding in ConvNorm, based on the relationship between cross-correlation and circular convolution established in Appendix B.2, we truncate the output of ConvNorm to make its spatial dimension align with the dimension of the input signal $z_{in}$. More specifically, after Step 3 in Appendix B.2, the resulting $\widetilde{z}_{out}$ has length $n + m - 1$,

---

[12] $\lfloor m \rfloor$ is the floor operation which outputs the greatest integer less than or equal to $m$.

in Step 4 we then truncate the first and last $(m-1)/2$ elements from it to make it has length $n$ as the input signal.

### B.4 Dealing with stride-2

Stride nowadays becomes an essential component in modern ConvNets [5,76]. A stride-$s$ convolution is a convolution with the kernel moving $s$ steps at a time instead of 1 step in a standard convolution shown in Figure 5. Mathematically, for the kernel $\boldsymbol{a}$ and the input $\boldsymbol{z}_{in}$, the stride-$s$ convolution can be written as

$$\boldsymbol{z}_{out} = \mathcal{D}_s\left[\boldsymbol{a} \circledast \boldsymbol{z}_{in}\right],$$

where $\mathcal{D}_s[\cdot]$ is a downsampling operator that selects every $s$th sample and discards the rest. Therefore, the main purpose of stride is for downsampling the output in ConvNets, replacing classical pooling methods. Hence for convolution with stride-$s$, the dimension of the output decreases by $s$ times in comparison to that of the standard stride-1 output. For example, if we do a stride-2 convolution on Figure 5 (ii), we will get the result $\boldsymbol{y} = [10]$ where the result is sampled from the standard stride-1 convolution output and its size is halved.

When enforcing weight regularizations, recent work often cannot handle strided convolution [29]. This happens because it involves weight matrix inversion, and the stride and the downsampling operator cause the weight matrix to be non-invertible. In contrast, since our method does not involve computing full matrix inversion and it operates on the outputs instead of directly changing the convolutional weights, we could first take a step back to perform an unstrided convolution, then use ConvNorm to normalize the output and finally do the stride (downsampling) operation on the normalized outputs.

### B.5 Dealing with 2D kernels

Although in the main body of the work, we introduced the ConvNorm based on 1D convolution for the simplicity of presentations, it should be noted that our approach can be easily extended to the 2D case via 2D FFT. For an illustration, let us consider (6), we know that in 1D case the preconditioning matrix for each channel can be written in the form of

$$\boldsymbol{P}_k = \left(\sum_{j=1}^{C_I} \boldsymbol{C}_{\boldsymbol{a}_{kj}} \boldsymbol{C}_{\boldsymbol{a}_{kj}}^\top\right)^{-\frac{1}{2}}$$

$$= \left(\sum_{j=1}^{C_I} \boldsymbol{F}^* \operatorname{diag}(\widehat{\boldsymbol{a}}_{kj}) \boldsymbol{F} \boldsymbol{F}^* \operatorname{diag}(\overline{\widehat{\boldsymbol{a}}}_{kj}) \boldsymbol{F}\right)^{-\frac{1}{2}} = \boldsymbol{F}^* \left(\left(\sum_{j=1}^{C_I} |\operatorname{diag}(\widehat{\boldsymbol{a}}_{kj})|^{\odot 2}\right)^{\odot-\frac{1}{2}}\right) \boldsymbol{F},$$

so that the output after ConvNorm can be rewritten as,

$$\boldsymbol{P}_k \boldsymbol{z}_k = \boldsymbol{F}^* \left(\sum_{j=1}^{C_I} |\operatorname{diag}(\widehat{\boldsymbol{a}}_{kj})|^{\odot 2})^{\odot-\frac{1}{2}}\right) \boldsymbol{F} \boldsymbol{z}_k = \boldsymbol{F}^{-1} \left(\sum_{j=1}^{C_I} |\boldsymbol{F}(\boldsymbol{a}_{kj})|^{\odot 2}\right)^{\odot-\frac{1}{2}} \boldsymbol{F}(\boldsymbol{z}_k),$$

where $\boldsymbol{F}(\cdot)$ and $\boldsymbol{F}^{-1}(\cdot)$ denote the 1D Fourier transform and the 1D inverse Fourier transform, respectively. To extend our method to the 2D case, we can simply replace the 1D Fourier transform in the above equation by the 2D Fourier transform. As summarized in Algorithm 1, to deal with 2D input data, we replace every 1D Fourier transform with 2D Fourier transform, which can be efficiently implemented via 2D FFT.

## C Experimental details for Section 4

In this part of appendix, we provide detailed descriptions for the choices of hyperparameters of baseline models, and introduce the settings for all experiments conducted in Section 4.

## C.1 Computing resources, assets license

We use two datasets for the demonstration purpose of this paper: CIFAR dataset is made available under the terms of the MIT license and ImageNet dataset is publicly available for free to researchers for non-commercial use. We refer the code of some work during various stages of our implementation for comparison and training purposes, we list them as follows: the implementation of ONI [39] is made available under the BSD-2-Clause license; the implementation of OCNN [43] is made available under the MIT license; the training procedure for Table 1 refers to the implementation of the work [62] which is made available under the MIT license and the black-box attack SimBA [56] implementation is made available under the MIT license. All experiments are conducted using RTX-8000 GPUs.

## C.2 Choice of hyperparameters for baseline methods

In Section 4, we compare our method with three representative normalization methods, that we describe the hyperparameter settings of each method below.

- **OCNN.** Since the best penalty constraint constant $\lambda$ for OCNN is not specified in [43], we do a hyperparameter tuning on the clean CIFAR-10 dataset for $\lambda \in \{0.001, 0.01, 0.05, 0.1, 1\}$ and picked $\lambda = 0.01$ from the best validation set accuracy.

- **ONI.** In the work [39], the authors utilize Newton's iteration to approximate the inverse of the covariance matrix for the reshaped weights. In our experiments, we adopt the implementation and use the default setting from their Github page where the maximum iteration number of Newton's method is set to 5. We use a learning rate 0.01 for all ONI experiments, where we notice that a large learning rate 0.1 makes the training loss explode to NaN.

- **SN.** In [15], the authors use the power method to estimate the spectral norm of the reshaped weight matrix and then utilize the spectral norm to rescale the weight tensors. For all SN experiments, we directly use the official PyTorch implementation of SN with the default settings where the iteration number is set to 1.[13]

## C.3 Experimental details for Section 4.1

**Robustness against adversarial attacks.** For gradient-based attacks, we follow the training procedure described in [62] to train models with our ConvNorm and other baseline methods.[14] Then we use Fast Gradient Sign Method (FGSM) [57] and Projected Gradient Method (PGD) [58] as metrics to measure the robust performance of the trained models. We note that the FGSM attack is defined to find adversarial examples in one iteration by:

$$\boldsymbol{x}_{adv} = \boldsymbol{x} + \epsilon * \text{sign}(\nabla_{\boldsymbol{x}}\ell(\boldsymbol{x}, \boldsymbol{y}, \boldsymbol{\theta}))$$

where $\boldsymbol{\theta}$ represents the model; $\boldsymbol{y}$ is the target for data $\boldsymbol{x}$ and $\epsilon$ denotes the attack amount of this iteration. PGD attack is an iterative version of FGSM with random noise perturbation as attack initialization. In this paper, we use PGD-$k$ to denote the total iterative steps (i.e., $k$ steps) for the attack methods. Adversarial attacks are always governed by a bound on the norm of the maximum possible perturbation, i.e., $\|\boldsymbol{x}_{adv} - \boldsymbol{x}\|_p \leq \delta_p$. We use $\ell_\infty$ norm to constrain the attacks throughout this paper (i.e., $p = \infty$). Specifically, we adopt the procedure in [62] by choosing $m = 4$ (the times of repeating training for each minibatch) and FGSM step $\epsilon = \frac{8}{255}$ during training. And we set the attack bound $\delta_\infty = \frac{8}{255}$.

For the black-box attack SimBA [56], we first train ResNet18 [5] models for ConvNorm and other baseline methods without BatchNorm on the clean CIFAR-10 training images using the default experimental setting mentioned in Section 4.[15] Then we choose the best model for each method

---

[13]The authors of SN take advantage of the fact that the change of weights from each gradient update step is small in the SGD case (and thus the change of the singular vector is small as well) and hence design the SN algorithm so that the approximated singular vector from the previous step is reused as the initial vector in the current step. They notice that 1 iteration is sufficient in the long run.

[14]For OCNN in the gradient-based attack experiment, we choose $\lambda = 0.0001$ since we found that this setting yields the best OCNN robust performance.

[15]We choose to not adding BatchNorm in the SimBA experiment because we empirically observe that removing BatchNorm improves the performance of every method.

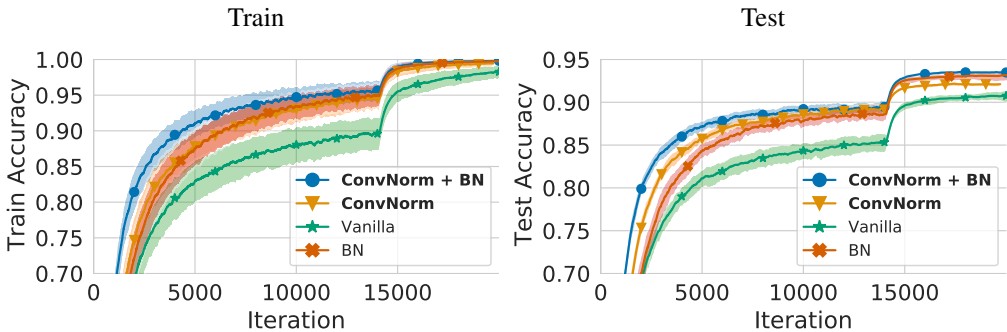

Figure 6: **Adding ConvNorm before BatchNorm accelerates convergence and improves performance.** Train and test accuracy of ResNet18 trained on CIFAR-10 with and without ConvNorm or BatchNorm under default settings mentioned in Section 4. Error bars correspond to min/max over 3 runs.

according to the best validation set accuracy. Finally, we apply each of the selected models on the held-out test set and randomly pick 1000 correctly classified test samples for running SimBA attack. We compare the performances of the selected models with pixel attack using a step size $\varepsilon = 0.4$. Since images in the dataset have spatial resolution $32 \times 32$ and 3 color channels, the attack runs in a total of $3 \times 32 \times 32 = 3072$ iterations. We then report the average queries and attack success rate after all 3072 iterations in Table 2.

**Robustness against label noise.** The label noise for CIFAR-10 is generated by randomly flipping the original labels. Here we show the specific definition. We inject the symmetric label noise to training and validation split of CIFAR-10 to simulate noisily labeled dataset. The symmetric label noise is as follows:

$$y = \begin{cases} y^{GT} \text{ with the probability of } 1 - r, \\ \text{random one-hot vector with the probability of } r, \end{cases}$$

where $r \in [0, 1]$ is the noise level. The models (using ResNet18 as backbones) are trained on noisily labeled training set (45000 examples) under the default experimental setting mentioned in Section 4. Max test accuracy is then reported on the held-out test set.

**Robustness against data scarcity** We randomly sample [10%, 30%, 50%, 70%] of the training data set CIFAR-10 dataset while keeping the amount of validation and test set amount unchanged. The model is trained on sub-sampled training set using the default experimental setting mentioned in Section 4. We report the accuracy on the held out test set by evaluating the best model selected on the validation set (obtained by randomly sampling 10% of the original training set).

### C.4 Generalization experiment and experimental details for Section 4.2

**Improved performances on supervised learning.** Below we provide the generalization experiment and more detailed experiment settings for fast training and generalization mentioned in Section 4.2. We use the default training setting mentioned in the **Setups of Dataset and Training** of Section 4 if not otherwise specified.

- *Faster training.* In order to isolate the effects of normalization techniques, we drop all regularization techniques including: data augmentations, weight decay, and learning rate decay as we have mentioned in the caption of Figure 4. For extra experiments on the same analysis when these regularization techniques are included, please refer to Appendix D and the results in Figure 6.

- *Better generalization.* We use the default setting as we have mentioned in the **Setups of Dataset and Training** of Section 4. All normalization methods are evaluated under this setting. We demonstrate the test accuracy of our method on CIFAR and ImageNet under standard settings. As shown in Table 4, although only using ConvNorm results in slightly worse test accuracy against BatchNorm, adding ConvNorm before standard BatchNorm layers can boost the performance while maintaining fast convergence (see Figure 6). Additionally, we investigate the influence of

| Dataset | Backbone | vanilla | BatchNorm(BN) | ConvNorm | ConvNorm + BN |
|---------|----------|---------|---------------|----------|---------------|
| CIFAR-10 | ResNet18 | $91.58 \pm 0.67$ | $93.18 \pm 0.16$ | $92.12 \pm 0.32$ | $\mathbf{93.31 \pm 0.17}$ |
| CIFAR-100 | ResNet18 | $66.59 \pm 0.72$ | $73.06 \pm 0.13$ | $68.20 \pm 0.27$ | $\mathbf{73.38 \pm 0.24}$ |
| ImageNet | ResNet18 | / | 69.76 | - | **70.34** |

Table 4: **Results on classification.** Test accuracy on CIFAR-10, CIFAR-100, and ImageNet validation sets. For each case, we compare different combinations of ConvNorm and BatchNorm. Results of CIFAR-10 and CIFAR-100 are averaged over 4 random seeds, and "/" represents failed training.

| | | Batch Norm | |
|---|---|---|---|
| | | ✓ | ✗ |
| Affine Transform | ✓ | $93.31 \pm 0.17$ | $92.12 \pm 0.32$ |
| | ✗ | $93.18 \pm 0.16$ | $92.01 \pm 0.21$ |

Table 5: **Ablation study.** The influence of the affine transform and batch normalization for classification on the CIFAR10 dataset is evaluated. The mean test accuracy and its standard deviation are computed over three random seeds.

| | Label Noise Ratio | | | |
|---|---|---|---|---|
| | 20% | 40% | 60% | 80% |
| ConvNorm + BN | $88.94 \pm 0.36$ | $85.88 \pm 0.26$ | $79.54 \pm 0.73$ | $69.26 \pm 0.59$ |
| ConvNorm | $87.75 \pm 0.13$ | $84.16 \pm 0.71$ | $77.48 \pm 0.26$ | $54.11 \pm 2.65$ |
| BN | $86.98 \pm 0.12$ | $81.88 \pm 0.29$ | $74.14 \pm 0.56$ | $53.82 \pm 1.04$ |
| Vanilla | $85.94 \pm 0.25$ | $82.11 \pm 0.52$ | $76.75 \pm 0.20$ | $57.20 \pm 0.71$ |

Table 6: **Adding ConvNorm and BatchNorm together makes a network more robust to label noise.** The influence of BatchNorm and ConvNorm for label noise on the CIFAR-10 dataset is evaluated. The mean test accuracy and its standard deviation are computed over three random seeds.

combining the affine transform and BatchNorm with ConvNorm. Table 5 shows the results of our ablation study on the CIFAR-10 dataset. Both affine transform and batch norm provide an independent performance boost.

**Training details for GAN** For GAN training, the parameter settings and model architectures for our method follow strictly with that in [15] and its official implementation for the training settings. More specifically, we use Adam ($\beta_1 = 0, \beta_2 = 0.9$) for the optimization with learning rate 0.0002. We update the discriminator 5 times per update of the generator. The batchsize is set to 64. We adopt two performance measures, Inception score and FID to evaluate the images produced by the trained generators. The ConvNorm is added after every convolution layer in the discriminator of GAN.

# D    Additional experiments and ablation study

In this section, we perform a more comprehensive ablation study to evaluate the influences of each additional component of ConvNorm on the tasks that we conducted in Section 4. More specifically, we study the benefits of the extra convolutional affine transform that we introduced in Section 3.3, as well as an inclusion of a BatchNorm layer right after the ConvNorm.

**Fast training and better generalization.** In Figure 4, we show that ConvNorm accelerates convergence and achieve better generalization performance with or without BatchNorm when regularizations such as data augmentation, weight decay, and learning rate decay are dropped during training. Figure 6 shows that when these standard regularization techniques are added, fast convergence of ConvNorm can still be observed (see the blue curve with circles and the yellow curve with triangles).

|  | Subset Percent | | | |
| --- | --- | --- | --- | --- |
|  | 10% | 30% | 50% | 70% |
| ConvNorm + BN | $77.96 \pm 0.11$ | $87.66 \pm 0.23$ | $90.49 \pm 0.17$ | $90.71 \pm 0.15$ |
| ConvNorm | $69.23 \pm 0.94$ | $83.93 \pm 0.34$ | $87.83 \pm 0.21$ | $89.85 \pm 0.10$ |
| BN | $67.10 \pm 2.59$ | $84.24 \pm 0.51$ | $88.74 \pm 0.73$ | $90.41 \pm 0.33$ |
| Vanilla | $67.56 \pm 0.50$ | $81.98 \pm 0.78$ | $86.57 \pm 0.35$ | $87.61 \pm 0.86$ |

Table 7: **Adding ConvNorm and BatchNorm together helps improve data efficiency** The influence of BatchNorm and ConvNorm for data scarcity on the CIFAR-10 dataset is evaluated. The mean test accuracy and its standard deviation are computed over three random seeds.

|  | Vanilla | ConvNorm | Cayley transfrom |
| --- | --- | --- | --- |
| Training time (epoch) | 21s | 60s | 182s |

Table 8: **Training time per epoch for different methods** The average training time for one epoch of different weight normalization methods is evaluated. Experiments are conducted CIFAR-10 dataset with a ResNet18 backbone.

**Robustness against label noise and data scarcity.** In Figure 3, we show that adding a BatchNorm layer after the ConvNorm can further boost the performance against label noise and data scarcity compared with combining other baseline methods with BatchNorm.

Here, to better understand the influence of each component, we study the effects of ConvNorm and BatchNorm separately. When we only use the ConvNorm without BatchNorm, from Table 6 and Table 7 we observe that in comparison to vanilla settings ConvNorm improves the performance against label noise and data scarcity for the most cases. In contrast, when only the BatchNorm is adopted, the performance downgrades that it improves upon the vanilla setting in some cases. Additionally, we notice that when we add $80\%$ of label noise to the training data, combining ConvNorm and BatchNorm together provides the best performance while using anyone alone would result in worse performance.

**Comparasion with Cayley Transfrom [29].** As mentioned in Section 1, a very recent work [29] shares some common ideas with our work in terms of exploring convolutional structures in the Fourier domain. We note that the major difference between [29] and our work lies in the trade-off between the degree of orthogonality enforced and the associated computational burden. As shown in Table 8, we empirically compare the run time for training one epoch of CIFAR-10 dataset on a ResNet18 backbone using different methods. We observe that both our method and [29] requires more time to train compared with the vanilla network. But since our ConvNorm explores channel-wise orthogonalization instead of layer-wise as done in [29], ConvNorm achieves faster training and [29] achieves more strict orthogonalization compared to each other. Also, we note that in terms of scalability, our ConvNorm could be adapted in larger networks such as ResNet50 and ResNet152, while the same experiments could not be carried on for [29] due to the limitation of our computational resources. Another important factor for comparison is the adversarial robustness. Based on our preliminary results, we found that ConvNorm has accuracy $46.12$ under PGD-10 attack, which outperforms the result of Cayley transform $38.35$ under the same attack. But we note that since the experiment settings in [29] are very different with ours, this comparison is not entirely fair as we have not done a comprehensive tuning for the Cayley transform method. We conjecture that with appropriate parameters and settings, the Cayley transform method could achieve on-par or even better results than ConvNorm since the more strict orthogonality enforced.

**Layer-wise condition number.** In Figure 2, we have shown that ConvNorm could improve the channel-wise condition number and layer-wise spectral norm. In this section, we empirically compare the layer-wise condition number of different normalization methods. We note that we use the method described in [51] to estimate the singular values and condition numbers of the actual convolution operators from each layer, not the weight matrix. Here, we define a metric $\rho$ to quantify the average

| | SN | ONI | OCNN | ConvNorm |
|---|---|---|---|---|
| $\rho$ | 2.724 | 0.001 | 2.288 | 3.332 |

Table 9: **Average layer-wise condition number ratio of vanilla method on top of other methods**
The experiments are conducted on natural settings with the same set of hyperparameter of Table 1.

ratio of the condition number of the vanilla method and other methods

$$\rho := \frac{1}{L} \sum_{l=1}^{L} \frac{\text{Condition number (Vanilla)}}{\text{Condition number (Method}_j}})$$

for characterizing the improvement upon the vanilla method (the larger, the better). From Table 9, we observe that our ConvNorm shows the best result in terms of improvement of layer-wise condition number as compared with other methods. We note that we did not compare with Cayley transform [29] because it inherently enforces more strict orthogonality than our ConvNorm based on their experiments, so we conjecture that Cayley transform could have better condition number than our ConvNorm.