# OpenReview forum: "Convolutional Normalization: Improving Deep Convolutional Network Robustness and Training"
_NeurIPS.cc/2021/Conference — NeurIPS 2021 Poster_

### Official Review · Reviewer_3mQk · 2021-07-02

**Rating:** 6
**Confidence:** 4

**Summary:**

The paper proposes to orthogonalize the convolutional kernel in a channel-wise manner. To do the computation more efficiently, they calculate the matrix inversion via FFT in the frequency domain. They compare with the baselines on the tasks of adversarial attack, image classification and GAN.

**Limitations And Societal Impact:**

The paper presents the limitations (e.g. theoretical explanation) and leaves them to future work.

**Main Review:**

Originality: I think that this paper is the first work proposed to orthogonalize output channels in the frequency domain via FFT. But this is not the first work that orthogonalizes weight in the frequency domain [1].

Clarity:

Pros: The paper is well-written and easy to follow overall.

Cons: Some typos:
1. F* is not defined in Eq. 3, though defined in Appendix.
2. 'Learnable affine transform' in line 235.

Quality:

Pros: The paper provides the comparison with multiple baselines on multiple tasks, and shows empirical advantages.

Cons:
1. I think the paper misses the empirical comparison with [1], at least in the task of adversarial attack. The proposed method shares a lot in common. E.g. focus on circular convolutions and Fourier domain, and [1] also shows the advantage in the task of adversarial attack.
2. It not clear how well the proposed method controls the spectral norm and condition number of the convolutional layer, and how well the input norm gets preserved, compared with baselines. Can you provide the spectral norm, condition number and ||Conv(x)||_2/||x||_2 (i.e. the ratio of the output norm and the input norm) for each layer, for ConvNorm and other baselines? (just in adversarial attack task is good enough)
3. I feel the run time is also not clear. Can you provide the run time comparison on ConvNorm and baselines under the different kernel settings (e.g. # channel_in, # channel_out, kernel_size, input_size)? I believe it can provide the community a better view of the advantages and limitations of each method.
4. The paper doesn't compare with orthogonalization-related baselines. And it seems the baselines have a better performance the ConvNorm. For instance, ONI [2] has 70.43 test acc on ImageNet with ResNet18, and OCNN [3] has 78.1 test acc on CIFAR-100 on ResNet18.
5. What's more, [2, 3] conducts the experiments with more complex architecture (e.g. ResNet 50, ResNet 101), while ConvNorm only shows the results on ResNet 18. So the scalability is my another concern.
6. I am confused about affine transform. It seems gamma_k is a freely learned kernel. I wonder if the orthogonality gets changed after passing affine transformation? How's the spectral norm, condition number gonna change?

Significance: I cannot determine the significance at this moment, since the paper doesn't justify itself completely to me. But I believe I will have a better and more overall understanding if more statistics and comparison can be provided. And I will consider raising my score then.


[1] Orthogonalizing Convolutional Layers with the Cayley Transform

[2] Controllable Orthogonalization in Training DNNs

[3] Orthogonal Convolutional Neural Networks

===============
post rebuttal: I have read the rebuttal; and raise my score accordingly.

**Time Spent Reviewing:**

6

---

> ### Author Response · Authors · 2021-08-10
> **Response to Reviewer 3mQk**
>
> We thank the reviewer for thoughtful and valuable comments. In the following, we address the reviewer’s comments one by one.
>
> ### 1. Comparison with [1’].
> We agree with the reviewer that the Cayley transform in [1’] and our work shares some common ideas in terms of exploring convolutional structures in the Fourier domain. As pointed out by the authors in [1’], one of the major open issues for the Cayley transform method is the heavy computation, mainly caused by the full matrix inversion. In comparison, we mitigate the computational issue by channel-wise orthogonalization, with a tradeoff in terms of layerwise orthogonality. We discuss this in more detail in Point 3.
>
> We apologize for missing the comparison with [1’] since we did not notice the GitHub of [1’] until very recently, which we will include in the updated version. During the rebuttal period, we ran and compared Cayley transform [1’] on adversarial robustness under PGD-10 attack, with the same setup as Table 1. Our initial investigation showed that our ConvNorm has accuracy 46.12 under PGD-10 attack, outperforming Cayley [1’] with 38.35 under the same attack. (we have not fine-tuned Cayley transform [1’] due to time constraints, it is highly possible that the Cayley transform [1’] would have better performance with careful hyperparameter tuning.) We also compared computational complexity, that we discuss in Point 3 with more details.
>
>
> ### 2. Controlling the spectral norm and condition number.
> In our paper, we have demonstrated substantial improvements on channel-wise condition number and layer-wise spectral norm (Figure 2). We also theoretically proved the control of layer-wise spectral norm (Prop. 3.1). Nonetheless, following the reviewer’s suggestions, during the rebuttal, we thoroughly estimated and compared the layer-wise spectral norm and condition number under the cleaning training setting (the estimations are similar in the adversarial training setting) with other methods. Our estimation of singular values is based on [4’], which calculates the singular values of the actual convolutional operators of each layer instead of estimation by treating the convolutional kernels as reshaped 2D matrices.
>
> First, we estimated the layer-wise spectral norm of the weight operator for each method on ResNet18 (i.e., averaging across layers), and report the following results:
>
> | | vanilla | SN [5'] | ONI [2’]| OCNN [3’] | ConvNorm |
> |--| ------- | ------- | ------- | ------- | -------|
> |Spectral Norm| 7.47    | 2.35    | 2.83    | 6.20 | 1.77 |
>
>
> we observe that our ConvNorm has the **smallest** layer-wise spectral norm.
>
> Second, for the layer-wise condition number, we report the average ratio $\rho$ of the condition number of the vanilla method to that of other methods (i.e., $\rho = \frac{condition\ number(Vanilla\ Method)}{condition\ number (method_{i})}$ ), where the **higher** the better. We obtain the results as:
>
> | | SN [5'] | ONI [2’]| OCNN [3’] | ConvNorm |
> | ------- | ------- | ------- | ------- | -------|
> | $\rho$    | 2.724    | 0.001    | 2.288 | 3.332 |
>
> We observe that our ConvNorm shows the best result in terms of layer-wise condition number as compared with other methods.
>
> Here we use the same hyperparameters as the experimental setup in Table 1. We did not compare with Cayley transform [1’] because it inherently enforces more strict orthogonality than our ConvNorm based on the experiments in [1’], it is supposed to have better condition number than our ConvNorm.
>
>
> ### 3. Runtime and scalability.
> We empirically compare the runtime of different methods on ResNet18 for CIFAR dataset by reporting the average elapsed time $t$ for one training epoch, we observe
>
> | | vanilla | ConvNorm | Cayley transform [1']|
> |--| ------- | ------- | ------- |
> |Training Time| 21s   | 60s   | 182s   |
>
> The runtime experiments are conducted on one NVIDIA V100 GPU. We observe that our ConvNorm is slower than the vanilla case while faster than the Cayley transform [1’], which corroborates our claim in Point 1. We notice that the runtime is specifically studied and nicely visualized in [1’]. We plan to incorporate similar experiments in our later versions of the manuscript. As for the scalability, our ConvNorm could be easily used in much deeper networks like ResNet152 for the CIFAR dataset (please refer to our response to *Reviewer HTRX*). We also conducted the same experiment in Table 1 on ResNet50 and noticed a similar performance pattern, which we will include in the updated manuscript. Additionally, for deeper networks such as ResNet50, we notice that the Cayley transform [1’] cannot be successfully trained on our end, which might be due to the limitation of our computational resources.
>
> ### 4. Comparison with other baseline methods.
> First, we would like to clarify that better generalization is not our main claim, rather, our method focuses on robustness and faster training. Nonetheless, for testing purposes, we compared OCNN on CIFAR-100 using the same setting as Table 4 in our paper, where the test accuracy for OCNN is $73.66$ and our ConvNorm achieves comparable test performance with $73.38$. In the revision, we will conduct more comprehensive experiments and include them in the appendix and discuss.
>
> ### 5. Explanation on affine transform.
> The reviewer is correct that $r_k$ is a freely learnable kernel. The intuition of adding the affine transform is analogous to that of the affine transform in BatchNorm (please refer to Section 3.3 in our paper for more detail). We believe that absolute orthogonality is not always desired in practice. Thus, we first orthogonalize the convolutional weights of a network and then adapt the network to the desired degree of orthogonality by the learnable affine transform. Typically, in the task of adversarial attack, we notice that the layer-wise spectral norms of convolutional operators are further suppressed after such affine transformations.
>
> ### References:
>
> [1’] Trockman, et al. Orthogonalizing Convolutional Layers with the Cayley Transform, ICLR’21.
>
> [2’] Huang, et al. Controllable Orthogonalization in Training DNNs, CVPR’20.
>
> [3’] Wang, et al. Orthogonal Convolutional Neural Networks, NeurIPS’19.
>
> [4’] Sedghi, et al. The Singular Values of Convolutional Layers, ICLR’19.
>
> [5’] Miyato, et al. Spectral Normalization for Generative Adversarial Networks, ICLR’18.

---

> > ### Comment · Reviewer_3mQk · 2021-08-24
> > **further questions**
> >
> > Thanks for the authors' replies, and they address most of my questions.
> > I have a few more questions about affine transform.
> > 1) Can you provide more details on how you calculate the "layer-wise spectral norms of convolutional operators"? I mean the "convolutional operator" means an orthogonal Conv layer and an affine transform altogether, or just an orthogonal Conv, or just an affine transform?
> > 2) Can you please provide the layer-wise condition numbers for it in the adversarial attack and image classification tasks?
> >
> > Thanks,
> >
> > Reviewer 3mQk

---

> > > ### Author Response · Authors · 2021-08-27
> > > **Re: further questions**
> > >
> > > We sincerely thank the reviewer for the appreciation of our responses, as well as great follow-up questions. We address the reviewer's questions as follows.
> > > ####  Q: Can you provide more details on how you calculate the "layer-wise spectral norms of convolutional operators"? I mean the "convolutional operator" means an orthogonal Conv layer and an affine transform altogether, or just an orthogonal Conv, or just an affine transform?
> > > A: Thanks for the good catch. First, our estimation of the condition number is based upon the work [4’], which estimates the condition number of the convolutional operator, rather than that of the reshaped weight matrix. Second, our estimation is based on the orthogonal Conv layer without the affine transform. Analogous to that of the BatchNorm, the affine transform could be initialized well conditioned (using delta kernel), and it would adapt to specific tasks during the training process. We will make this clear in the revision.
> > > #### Q: Can you please provide the layer-wise condition numbers for it in the adversarial attack and image classification tasks?
> > > A: The reviewer raised a very good point. In the natural training setting, the ratio $\rho$ between the condition number of the vanilla model and the model with ConvNorm on the image classification task (CIFAR10) is 3.332 as we reported in the Table above. For the adversarial setting, the ratio $\rho$ is 2.087.  We note that in both cases, our method obtains a better condition number than the vanilla model. We will incorporate the results in the experimental part of the revision as well.

---

### Official Review · Reviewer_G1ti · 2021-07-04

**Rating:** 6
**Confidence:** 3

**Summary:**

The paper introduces a new way to enforce orthogonality in convolutional layers. The main idea is to make the computations in the frequency domain where convolutions (or cross-correlations) become products. Hence, exact orthogonality is achieved, while keeping the computational costs low using FFT.

**Limitations And Societal Impact:**

Yes

**Main Review:**

Major revision

The paper is well written, although some of the computations are not easy to follow. Section 3.1 could be enlarged and better explained. If the authors need more space to do that, the previous section regarding neural networks in general could be deleted. It is only useful because it provides some notation.

The forward pass of the new layer is not very clear in the paper. It could be useful to explain it, because it might seem that you provide a formula for $Q(a)$ and then multiply the $x$ by that matrix, which would require a large computation and which you do not do. It gets more clear in the Appendix, but it is worth mentioning it in the paper. Besides, experiments of the computation time of the forward pass and a comparison with other methods would be valuable in the paper, since you claim that your method is faster and show that analytically.

Minor revisions:

Line 157: "The basic idea is to multiply a preconditioning matrix which approximates the following form": the sentence is not very clear in a first reading, now I understand what you mean, but please rephrase.
Line 177: "However, generalizing our method from a single channel to multiple channels is not obvious, that we discuss below.": again, please rephrase this sentence.

Overall Evaluation:
Apart from simple changes that can be made by the authors, the idea of the authors is clever and the implementation efficient. Besides, numerical experiments prove that their approach is valuable. At the moment, I think the paper is a little above the acceptance threshold and could be further improved with experiments regarding inference time.


**Time Spent Reviewing:**

10

---

> ### Author Response · Authors · 2021-08-10
> **Response to Reviewer G1ti**
>
> We thank the reviewer for thoughtful and valuable comments. In the following, we address the reviewer’s comments in detail.
>
> ### 1. Rewrite Section 3.1.
> We thank the reviewer for the suggestion. In the revision, we will rewrite Section 3.1 by incorporating more technical details (from Appendix B), and shorten Section 2 correspondingly. We will mention the corresponding appendix section (e.g. Appendix B) in the main text in the revised version.
>
>
> ### 2. More detailed explanations on the forward pass.
> As we have described in Algorithm 1 of the appendix, for each layer, the forward pass of ConvNorm consists of three parts: the activation from the previous layer would go through (i) the convolutional layer and then (ii) the proposed ConvNorm, followed by (iii) passing through the nonlinear activation layer ReLU. The proposed ConvNorm can be implemented efficiently in the frequency domain, by Fourier transform and conducting element-wise multiplications. In the revision, we will incorporate this explanation and the pipeline of Algorithm 1 in the main body. Compared with the vanilla ConvNets, because of the efficiency of FFTs, the increase of forward computational time is not much as we show in the following.
>
> ### 3. Computational complexity.
> When doing inference, for forward propagation, our implementation will directly use $\hat{\bf v}_k$ (as in Algorithm 1) learned and stored during training. Thus, we only need to do FFT and inverse FFT once on each convolutional layer’s input and output. This also shortens the inference time of our method. For comparison, the inference time $t$ of a vanilla ResNet18 network in terms of one epoch of CIFAR10 testset (i.e., 10000 samples) is on average $1.594 s$, where the inference time of our method under the same setting is $4.699 s$. In the revision, we will include this in the experiment part and discuss it. For other comparisons of the training time, we refer the reviewer to our response to *Reviewer HTRX* (Point 1).
>
> ### 4. Addressing minor comments.
> In the revision, we will rephrase the sentences in Line 157 and Line 177 as suggested to make the meaning more clear to the readers.

---

### Official Review · Reviewer_HTRX · 2021-07-11

**Rating:** 7
**Confidence:** 4

**Summary:**

This paper exploits the convolutional structure to maintain orthogonality on the convolutional kernel for guarantee the energy preservation in CNN, rather than the naively used methods using flattened convolutional kernels. It orthogonalizes the channel-wise convolutional kernel via preconditioning its circulant matrix transferred by FFT. The proposed ConvNorm also reduces the layer-wise spectral norm of the weight matrices. Experiments on small network (e.g, VGG 16 and 18-layer Resnet) using CIFAR-10 ImageNet verify the effectiveness of ConvNorm in model robustness and training GANs.

**Limitations And Societal Impact:**

Yes

**Main Review:**

**Pros:**

+The idea to orthogonalize the channel-wise convolutional kernel via preconditioning its circulant matrix is novel to me.
+The related works are comprehensive discussed, as far as I am concerned.
+The presentation is well organized
+The effectiveness of ConvNorm in model robustness and training GANs are well validated by the experiments.



**Cons:**

1.One of my main concern is the computational or memory cost of the proposed methods. This paper spends much of contents to show the computational problems of the previous methods (Lines 58~61, Line 76) and claims the efficiency of the proposed ConvNorm. However, I did not see analysis about the computational complexity or the time cost in experiments. What is the wall-clock time cost of the proposed method, compared to the baselines? I would like to see the time cost& memory cost in the described experiments, especially on ImageNet.
2. I am not convinced with the claims“Easier training and better generalization” by the experiments: 1) Clearly, ConvNorm do not work well without BN in such a small network (VGG16-ResNet18); 2) the experiments should be conduct on larger network, e.g, 56-layer or 110-layer ResNet on CIFAR (1GPU is enough to run the experiments) or better using ResNet-50 on ImageNet. I wonder whether ConvNorm can work on 110-layer ResNet without BN.

**Other comments:**

What is the y-axis of the Figure 2 (Left), the log-scale? It should be described in here. By the way, I believe it is more important to show the layer-wise condition number of the proposed ConvNorm, rather than the channel-wise condition number, because the motivation of orthogonal matrix W is for preserving the magnitude between the output and input, with y=Wx, or $\frac{\partial{L}}{\partial{x}} = W^T \frac{\partial{L}}{\partial{y}}$.

It is not clear why should stop the gradient shown in Line 229-234.  I believe this can work with feature activation (e.g. BN) in a network, but I conjecture it will suffer training instability without feature activations in deeper networks (e.g., 56-layer/110-layer ResNet for CIFAR classification). It is better to provide the results in deeper networks without feature activation.

**update**:
I have read authors’ responses and other reviewers’ comments. I acknowledge the authors’ feedback which addresses my concerns. I hope the code (including the additional experiments for comparing layer-wise condition number) will be released if this paper gets accepted. I keep my score as 7.


**Time Spent Reviewing:**

5 hours

---

> ### Author Response · Authors · 2021-08-10
> **Response to Reviewer HTRX**
>
> We thank the reviewer for thoughtful and valuable comments.
>
> First of all, as raised by *Reviewer 3mQk*, it should be noted that our method has not been compared with the very recent Cayley transform method [1’], because we did not notice the GitHub of [1’] until very recently. During the rebuttal, because of the similarities between the two methods, we tested and compared with [1’]. Please refer to our response to *Reviewer 3mQk* for more details.
>
> During the rebuttal, we have conducted more comprehensive experiments as the reviewer suggested. We address the reviewer’s comments in detail as follows.
>
> ### 1. Computational complexity.
> We empirically compare the runtime of different methods by reporting the average elapsed time $t$ for a training epoch (CIFAR dataset) on ResNet18. The result is the following:
>
> | | vanilla | ConvNorm | Cayley transform [1'] |
> |--| ------- | ------- | -----------|
> |Training Time| 21s   | 60s   | 182s |
>
> The runtime experiments are conducted on one NVIDIA V100 GPU. We observe that our ConvNorm is slower than the baseline methods while faster than the Cayley transform method [1’].
>
> ### 2. Clarification on “easier training and better generalization’’ and scalability.
> First, we want to clarify that better generalization is not our main claim, rather, our method focuses on robustness and faster training. Nonetheless, we agree with the reviewer that it is a good idea to include more comprehensive experiments on deeper networks to demonstrate the scalability of our method. During the rebuttal, we ran several experiments on networks of different depths without BN to examine the scalability of ConvNorm. We found that our ConvNorm could be easily used on very deep networks for the CIFAR dataset. For example, we can easily train a ResNet152 on CIFAR100 without BN, where our ConvNorm achieves an accuracy of $70.29$ in comparison to the baseline (vanilla method without BN) with an accuracy of $67.65$.
>
> ### 3. Layer-wise condition number.
> During the rebuttal period, we have conducted more comprehensive experiments on comparing layer-wise spectral norm and condition number. Our estimation of singular values is based on [4’], which calculates the singular values of the actual convolutional operators of each layer instead of estimation by treating the convolutional kernels as reshaped 2D matrices.
>
> First, we estimated the layer-wise spectral norm of the weight operator for each method on ResNet18 (i.e., averaging across layers), and report the following results:
>
> | | vanilla | SN [5'] | ONI [2’]| OCNN [3’] | ConvNorm |
> |--| ------- | ------- | ------- | ------- | -------|
> |Spectral Norm| 7.47    | 2.35    | 2.83    | 6.20 | 1.77 |
>
> We observe that our ConvNorm has the **smallest** layer-wise spectral norm.
>
> Second, for the layer-wise condition number, we report the average ratio $\rho$ of the condition number of the vanilla method to that of other methods (i.e., $\rho = \frac{condition\ number(Vanilla\ Method)}{condition\ number (method_{i})}$ ), where the **higher** the better. We observe:
>
> | | SN [5'] | ONI [2’]| OCNN [3’] | ConvNorm |
> | ------- | ------- | ------- | ------- | -------|
> | $\rho$    | 2.724    | 0.001    | 2.288 | 3.332 |
>
> We observe that our ConvNorm shows the best result in terms of layer-wise condition number as compared with other methods.
>
> Here we use the same hyperparameters as the experimental setup in Table 1. We did not compare with Cayley transform [1’] because it inherently enforces more strict orthogonality than our ConvNorm based on the experiments in [1’], it is supposed to have better condition number than our ConvNorm. However, as shown in Point 1, the Cayley transform method [1’] is much slower than ours.
>
> ### 4. Stop gradient in Line 229-234 and training stability.
> In experiments, we tested the method with and without the stop gradient scheme. They show minimal differences in terms of overall performance. However, using stop gradients will be faster and easier to implement. In terms of training stability, we conducted extra experiments, observing that deeper networks (e.g., ResNet152) with ConvNorm and stop gradient can still be efficiently trained without BatchNorm. We will explain and discuss these in the revision, and add comparison experiments.
>
> ### 5. Other minor comments.
> We will add the description of the y-axis of Figure 2 (Left) in the revision, which is the natural scale instead of the log scale.
>
> ### References:
>
> [1’] Trockman, et al. Orthogonalizing Convolutional Layers with the Cayley Transform, ICLR’21.
>
> [2’] Huang, et al. Controllable Orthogonalization in Training DNNs, CVPR’20.
>
> [3’] Wang, et al. Orthogonal Convolutional Neural Networks, NeurIPS’19.
>
> [4’] Sedghi, et al. The Singular Values of Convolutional Layers, ICLR’19.
>
> [5’] Miyato, et al. Spectral Normalization for Generative Adversarial Networks, ICLR’18.

---

> > ### Comment · Reviewer_HTRX · 2021-08-20
> > **Concerns on the layer-wise condition number**
> >
> > I thank for the authors' feedbacks, which addresses most of my concerns. I still have one concern on the result of the additional  layer-wise condition number: It is weired why SN has 2.724, while ONI has 0.001? Does here SN use the same way for convolution as ONI (e.g.  by treating the convolutional kernels as reshaped 2D matrices)?   I believe ONI will also has theoretical better conditioning than SN ( Here, I have noted the authors calculate the singular values of the actual convolutional operators of each layer instead of estimation by treating the convolutional kernels as reshaped 2D matrices.)
> >
> > Bests,
> >
> > Reviewer HTRX

---

> > > ### Author Response · Authors · 2021-08-22
> > > **Response to reviewer's concern on the layer-wise condition number**
> > >
> > > We thank the reviewer for the appreciation of our response and great follow-up questions. For the reported poor condition number of ONI, we also think the underlying reason is that ONI is proposed for improving the condition number of the reshaped weight matrix rather than that of the convolutional operator. To make sure our implementation of ONI is correct, we double-checked the following:
> > > - First, we tested the condition number of reshaped weight matrix for ONI, which is $2.15$ compared to $4.39$ (learning rate $0.01$) of the vanilla model. This suggests that ONI indeed reduces the condition number of the reshaped weight matrix.
> > > - Second, the learning rate we used for ONI in the previous response is $0.01$, which is different from that $0.1$ for training all the other baseline methods. This is because we found ONI cannot be successfully trained on our end when the learning rate is $0.1$. In this follow-up, we compared ONI with the vanilla model both trained with the same smaller learning rate $0.01$, still, ONI shows a poor condition number than the vanilla case.
> > >
> > > Admittedly, we just used the preset hyperparameters for ONI and have not fine-tuned them. Nonetheless, our preliminary investigation supports our conclusion. We think the underlying reason is that ONI is modifying the singular values of the reshaped weight matrix nonuniformly, which substantially influences the condition number of the convolutional operator. In comparison, since SN divides all singular values uniformly by the max singular value, thus SN should not have any direct impact on the condition number.

---

### Official Review · Reviewer_FWBT · 2021-07-17

**Rating:** 7
**Confidence:** 4

**Summary:**

The paper proposes a novel normalization technique for controlling the channel-wise Lipschitz of the convolutional layers. The normalization technique operates on the Fourier domain which could reduce the burden of doing full SVD and at the same time be more accurate compared to using approximate methods. The paper also empirically demonstrates the effectiveness of such normalization by showing experimental results on robustness under different sources of noises, easier training, and extended usage in GAN.


**Limitations And Societal Impact:**

Yes

**Main Review:**

I'm quite positive in general. The paper is clearly motivated, well written, and smoothly presented. The method proposed is reasonable and well established. Also, the implementation is straightforward and since there is no need to tune hyperparameters, the method could easily serve as a plug-in module for most deep convolutional networks. The experimental part is comprehensive, the authors tried various sets of tasks to empirically test the performances of the proposed method and other methods.

One remaining concern is that most experiments in this paper are primarily done on CIFAR-10, it would be more interesting to see more experiments conducted on more complex datasets like Imagenet.

**Time Spent Reviewing:**

1

---

> ### Author Response · Authors · 2021-08-10
> **Response to Reviewer FWBT**
>
> We very much thank the reviewer for the appreciation of our work. While some experiments (such as with noisy labels) are conducted on CIFAR10 for the purpose of illustration, we do verify the effectiveness of the proposed convolutional normalization on ImageNet (like Fig. 4 and Table 1 in the Appendix).  Following the reviewer’s suggestion, in the revision, we will include more experiments on ImageNet and other datasets.

---

### Decision · Program_Chairs · 2021-09-27

**Decision:**

Accept (Poster)

**Comment:**

In this paper, the authors present a simple and efficient ConvNorm method that can fully exploit the convolutional structure in the Fourier domain. The paper is clearly written and well motivated. Extensive experiments empirically demonstrate the effectiveness of such normalization. As all reviewers achieve consensus that the paper is well written and can be accepted, I vote for acceptance. The authors are expected to make a thorough revision by considering the the reviews.